**COMMUNICATIONS**

# NMT1 and NMT2 are lysine myristoyltransferases regulating the ARF6 GTPase cycle

Tatsiana Kosciuk [1], Ian R. Price [1], Xiaoyu Zhang[1], Chengliang Zhu[1], Kayla N. Johnson[1], Shuai Zhang[1,2], Steve L. Halaby[3], Garrison P. Komaniecki[1], Min Yang[1], Caroline J. DeHart[4], Paul M. Thomas [4], Neil L. Kelleher [4], J. Christopher Fromme [3] & Hening Lin [1,2]✉

Lysine fatty acylation in mammalian cells was discovered nearly three decades ago, yet the enzymes catalyzing it remain unknown. Unexpectedly, we find that human N-terminal glycine myristoyltransferases (NMT) 1 and 2 can efficiently myristoylate specific lysine residues. They modify ADP-ribosylation factor 6 (ARF6) on lysine 3 allowing it to remain on membranes during the GTPase cycle. We demonstrate that the NAD$^+$-dependent deacylase SIRT2 removes the myristoyl group, and our evidence suggests that NMT prefers the GTP-bound while SIRT2 prefers the GDP-bound ARF6. This allows the lysine myrisotylation-demyristoylation cycle to couple to and promote the GTPase cycle of ARF6. Our study provides an explanation for the puzzling dissimilarity of ARF6 to other ARFs and suggests the existence of other substrates regulated by this previously unknown function of NMT. Furthermore, we identified a NMT/SIRT2-ARF6 regulatory axis, which may offer new ways to treat human diseases.

[1] Department of Chemistry and Chemical Biology, Cornell University, Ithaca, NY 14853, USA. [2] Howard Hughes Medical Institute; Department of Chemistry and Chemical Biology, Cornell University, Ithaca, NY 14853, USA. [3] Department of Molecular Biology and Genetics; Weill Institute for Cell and Molecular Biology, Cornell University, Ithaca, NY 14853, USA. [4] National Resource for Translational and Developmental Proteomics, Departments of Chemistry and Molecular Biosciences and the Feinberg School of Medicine, Northwestern University, Evanston, IL 60208, USA. ✉email: hl379@cornell.edu

Lysine fatty acylation was recently identified on several Ras small GTPases and was found to regulate their cellular localization and activity[1–3]. Members of the sirtuin family of nicotinamide adenine dinucleotide (NAD$^+$)-dependent deacylases are known erasers of lysine fatty acylation in mammalian cells[1–5]. Although several bacterial toxins have been reported to catalyze lysine fatty acylation[6,7], such enzymes in mammals have not been found despite the long-known occurrence of lysine fatty acylation in uninfected cells[8,9].

N-terminal glycine myristoylation is catalyzed by the N-myristoyltransferases (NMT) after the initiator methionine is removed by methionine aminopeptidase[10]. There are two human NMT enzymes and they have been explored as therapeutic targets for malaria[11], sleeping sickness[12–14], common cold[15], and cancer[16–18]. N-terminal glycine myristoylation is required for the reversible membrane association of ARF (ADP-ribosylation factor) proteins[19,20]. ARFs are small GTPases that cycle between the active GTP-bound and the inactive GDP-bound states. The GTP hydrolysis is facilitated by GTPase-activating proteins (GAPs) followed by guanine nucleotide exchange factor (GEF)-mediated exchange of GDP to GTP. In the GTP-bound state, the amphipathic N-terminal helix with the myristoyl group is inserted into membranes and ARFs bind their effectors to regulate essential trafficking and signaling pathways. Owing to the conformational change caused by GTP hydrolysis, the myristoylated amphipathic helix is sequestered in a hydrophobic pocket allowing ARFs to dissociate from membranes and effectors leading to attenuation of signaling[21]. ARFs 1–5 reside at the Golgi and regulate the Golgi–endoplasmic reticulum traffic. ARF6, however, localizes to the plasma membrane and the endocytic system and, unlike other ARFs, tends to remain membrane bound even in the inactive state[21,22]. This has been a puzzle as ARF6 has a high structural similarity to ARF1 and follows the same nucleotide-dependent dynamics of the amphipathic helix[23,24]. Unexpectedly, we found that human NMT1 and NMT2 can catalyze lysine myristoylation of ARF6 providing an explanation for its unusual membrane association.

## Results

**NMT1 and NMT2 act on lysine residues in vitro**. NMTs have a strong preference for the peptide sequence GXXXS[20,25]. It has been shown that the selectivity for glycine is due to the ability of the α-amine of the N-terminal glycine to rotate and attack the carbonyl carbon of myristoyl-CoA in the active site of NMT without the steric hindrance that would be experienced by other residues[26]. We reasoned that the ε-amine of lysine could sterically mimic the α-amine of the N-terminal glycine and therefore might react in the active site of NMT (Fig. 1a). Given this, we thought that, if the lysine reaction is possible, the NMT substrate sequence requirements for lysine and N-terminal glycine modifications would also be similar. Therefore, we predicted that the preferred sequence for lysine myristoylation would be XKXXS or KXXS. The N-terminus of ARF6, the known substrate of NMT, fits this motif with a lysine residue (K3) following the N-terminal glycine (Fig. 1b). Thus we hypothesized that ARF6 could be an NMT lysine myristoylation substrate.

We first tested this possibility in vitro using purified recombinant NMT and synthetic peptides derived from the N-terminus of ARF6. To block the NMT activity toward the N-terminal glycine (G2), we deleted G2 leaving lysine at the N-terminus, or changed G2 to alanine (G2A), or acetylated the amino group of G2 (Ac-G). Excitingly, both NMT1 and NMT2 were able to modify all these peptides (Fig. 1c). With the KVLSIF peptide, the reaction still occurred when the α-amino group of K was acetylated, but no reaction occurred when K was switched to

R (Fig. 1c, Supplementary Fig. 1A, B, F), confirming that the lysine side chain, but not the N-terminal α-amino group or other residues, was modified. The modification of the lysine on the peptides was further confirmed by tandem mass spectrometry (MS/MS; Supplementary Fig. 2). Kinetics comparison revealed that the modification of the G2A peptide proceeded with an overall efficiency of about half to a third of that for K3R (Supplementary Fig. 1C), suggesting that K3 lysine myristoylation could occur rather efficiently.

Next, we tested whether the ARF6 N-terminal peptide with available N-terminal glycine could be modified on lysine 3. We used synthetic peptide standards myristoylated on lysine or glycine to determine the identity of the formed species using high-performance liquid chromatography (HPLC). Owing to high peptide hydrophobicity, we were unable to obtain the doubly myristoylated peptide standard. Under the reaction conditions used, mostly glycine myristoylation occurred and the lysine myristoylation product peak was very small (Fig. 1d), suggesting that under these conditions, glycine myristoylation was more efficient on peptide substrates.

NMT is thought to predominantly act cotranslationally; however, there is mounting evidence for its posttranslational activity[27]. We therefore asked whether the substrate three-dimensional structure could affect lysine myristoylation. We expressed and affinity-purified Flag-tagged ARF6 and its G2A, K3R, and G2A/K3R mutants from cells treated with a dual NMT1/NMT2 inhibitor DDD85646. We then incubated the purified proteins with recombinant NMT1 or NMT2 in the presence of Alk12-CoA, an alkyne-tagged myristoyl-CoA analog. An azide-containing fluorescent dye was conjugated to the alkyne tag via click chemistry and the labeling was analyzed by in-gel fluorescence. Consistent with the synthetic peptide results, both NMTs could modify ARF6 wild-type (WT), G2A, and K3R mutants, but not G2A/K3R, supporting that NMT can myristoylate K3 of ARF6 in vitro (Fig. 1e and Supplementary Fig. 1D). Interestingly, the reaction on the ARF6 WT produced a laddering fluorescence and immunoblot band patterns, suggesting that there were several populations of myristoylated ARF6 proteins, likely due to myristoylation on glycine or lysine (1-myr), or both (2-myr). Later, we provided more evidence that one of the 1-myr bands is indeed lysine myristoylated ARF6. We therefore concluded that the three-dimensional structure of the substrate might influence the NMT lysine myristoylation activity.

To further understand the new activity of NMT, we obtained X-ray crystal structures of NMT in complex with lysine peptide substrates. Co-crystallization of NMT2 with a KVLSKIF peptide and myristoyl-CoA resulted in a 1.93 Å structure capturing the myristoyl lysine peptide product, clearly showing that the lysine ε-amine, but not the α-amine, was modified (Supplementary Fig. 6A). Furthermore, the simulated-annealing omit map unambiguously demonstrates the covalent bond between the lysine and myristoyl, concomitant with the loss of electron density connecting the myristoyl group and CoA (Fig. 1f). The comparison of our NMT2 structure with previously determined structures of NMT1 bound to myristoylated peptide products[25] revealed that lysine binds analogously to the glycine substrate, with the amide bond directly overlapping between the structures (Fig. 1h).

Interestingly, there was little electron density for the adenosine-3′-phosphate of the CoA bound to NMT2, suggesting that it was either very flexible or hydrolyzed at the 5′ phosphate. Including or omitting this moiety made little difference in the R-free value, so it was removed from the final model (Fig. 1f, Supplementary Fig. 6A, and Supplementary Table 1). Similarly, the region from Arg115 to His135, involved in binding to the adenosine,

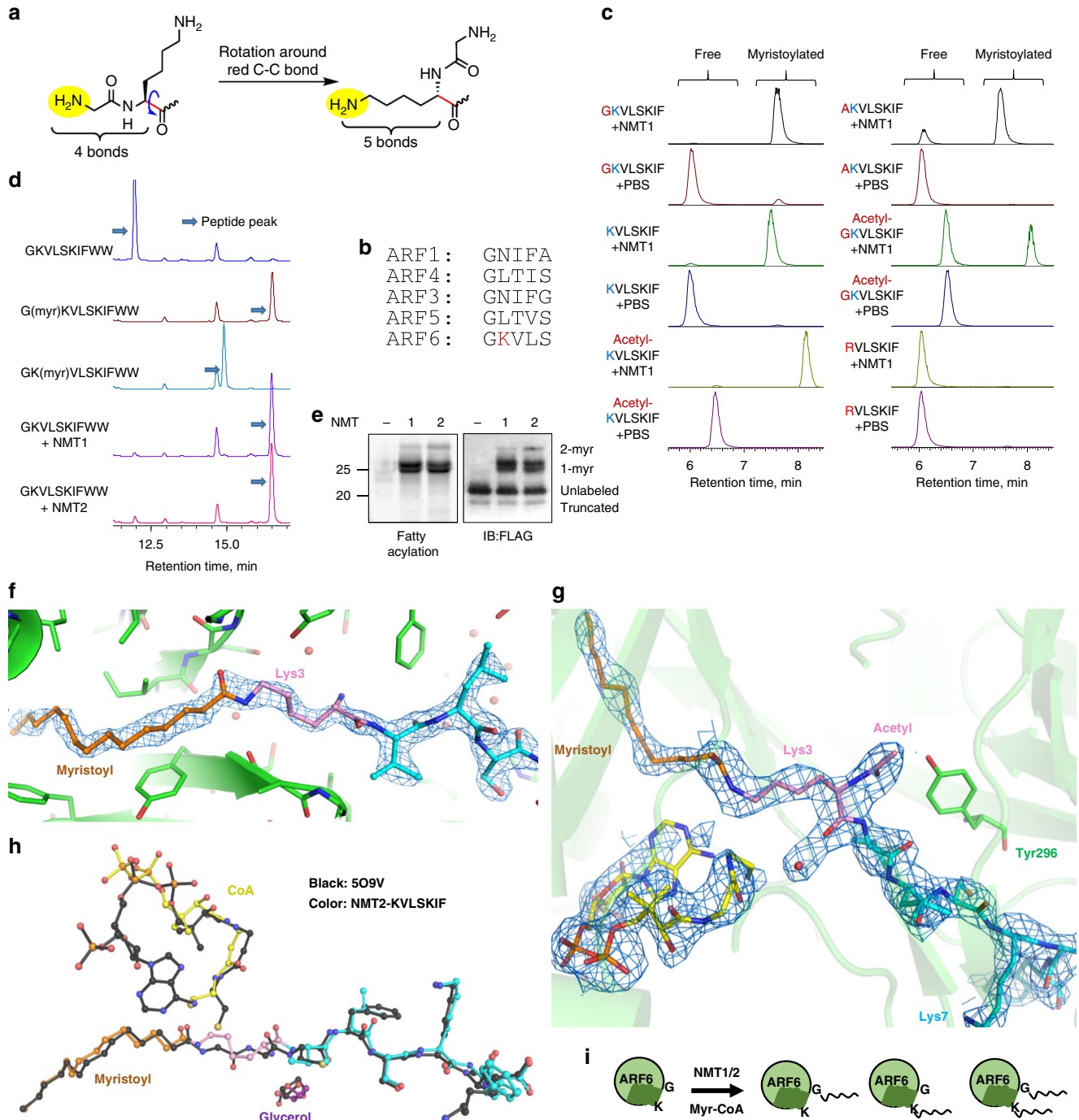

**Fig. 1 NMT1 and NMT2 have lysine transferase activity and can modify ARF6 on K3. a** Lysine steric properties resemble those of N-terminal glycine. **b** The alignment of the N-terminal sequences of human ARFs reveals a unique lysine residue in ARF6. **c** Monitoring NMT1 in vitro reactions on different ARF6 peptides using LC-MS. Total ion chromatograms searched for the substrate and product ions are shown. **d** HPLC separation of NMT reaction on ARF6 WT N-terminal peptide reveals mostly glycine myristoylated product. **e** NMT reaction with ARF6 WT protein and Alk12-CoA generates several modified species. Shown are in-gel fluorescence and FLAG western blot after TAMRA azide conjugation via click chemistry. **f** Simulated annealing omit $2F_O$–$F_C$ map (at 1.2 σ) surrounding the myristoyl-KVLSKIF product in NMT2. Peptide: cyan, Lys3: pink, myristoyl: orange. **g** Myristoyl–peptide and CoA products bound to chain A in the NMT1 myristoyl-AcKVLSKIF structure. The 1.0 σ $2F_O$–$F_C$ electron density map around the ligands is shown (blue mesh). **h** Comparison with PDB structure 5O9V (black), NMT1 containing myristoyl-glycine peptide product. **i** Model showing that NMT1 and NMT2 enzymes can modify G2, K3 of ARF6, or both.

was also flexible and thus not modeled, unlike in previous NMT structures.

We then determined a 2.5-Å crystal structure of NMT1 with an Ac-KVLSKIF peptide, in which the N-terminus (α-amine) is acetylated and thus cannot be myristoylated. Analogous to the higher-resolution NMT2 structure with KVLSKIF peptide, we still observed continuous electron density between the lysine and the myristoyl, with a loss of electron density connecting the CoA sulfur to the myristoyl (Fig. 1g and Supplementary Fig. 6B). The carbonyl of the acetyl N-terminus forms a hydrogen bond with

Tyr296, similarly to the acetyl in PDB structure 5O9T[25]. This structure confirms that the lysine residue, not the N-terminus, is modified.

**NMT catalyzes ARF6 lysine myristoylation in cells**. To test whether lysine myristoylation by NMT can occur in live cells, we treated HEK293T cells transiently expressing ARF6 G2A that could only be myristoylated on lysine with Alk12 or Alk14, clickable myristic and palmitic acid analogs, with or without the NMT inhibitor, and the fatty acylation levels were analyzed as outlined in Fig. 2a. Since NMT prefers myristoyl-CoA over palmitoyl-CoA, we speculated that Alk12 labeling would be more efficient than Alk14[28] and would be abolished by pharmacological NMT inhibition, if NMT was the transferase. We indeed observed the predicted effect (Fig. 2b), supporting that NMT can myristoylate ARF6 G2A in cells. We further confirmed that myristoylation occurred on K3 of ARF6 G2A protein isolated from HEK293T cells by MS (Supplementary Fig. 3).

To examine whether NMT knockdown (KD) could decrease lysine myristoylation of ARF6, we expressed ARF6 G2A in NMT1 or NMT2 KD HEK293T cells (Supplementary Fig. 15). Only NMT1 KD decreased the myristoylation levels of ARF6 G2A, which could be rescued by the overexpression (OE) of HA-NMT1 or HA-NMT2 (Fig. 2c). Therefore both NMT enzymes can act on K3 of ARF6, but NMT1 is likely the major endogenous myristoyltransferase in HEK293T cells under our experimental conditions. Together these data suggest that NMT1 and NMT2 can myristoylate both G2 and K3 of ARF6.

Given that in vitro reaction with ARF6 and NMT produced dimyristoylated ARF6 (Fig. 1e and Supplementary Fig. 1D), we asked whether di-myristoylated ARF6 could also be produced in cells. To test that we expressed ARF6 WT, G2A, K3R, and G2A/K3R mutants in HEK293T cells along with NMT1 or NMT2 and checked the Alk12 labeling. All variants except G2A/K3R produced fluorescent signal that was increased by NMT expression. However, for WT ARF6 and only for WT ARF6, NMT1 and NMT2 expression produced a higher molecular weight band (Fig. 2d) similar to that in the in vitro labeling of purified proteins (Fig. 1e and Supplementary Fig. 1D), suggesting that ARF6 could have both monomyristoylation and dimyristoylation in cells. Interestingly, the dimyristoylated band produced by NMT2 OE in cells was more pronounced compared to that from in vitro acylation (compare Fig. 2d to Fig. 1e), which suggests that other factors such as cellular localization or binding partners might regulate ARF6 dimyristoylation by NMT. In addition, NMT2 OE produced more dimyrisotylated ARF6 than NMT1 OE (Fig. 2d), but in vitro there was little difference between the two enzymes (Fig. 1e and Supplementary Fig. 1D), suggesting an additional level of regulation of this activity in cells. Since NMT2 produced more lysine myristoylated product than NMT1, we used NMT2 to further study ARF6 lysine myristoylation and to confirm dimyristoylation on G2 and K3 by top–down MS (Fig. 2e and Supplementary Fig. 4). Based on the quantification of the total ion chromatogram peaks, the dimyristoylated species is relatively abundant, about one third of the modified pool (Fig. 2e). Top–down MS analysis also revealed a truncated ARF6 (Fig. 2e) generated from an alternative start site, which explains the presence of the lowest unlabeled FLAG immunoblot band as those in Figs. 1e and 2c.

Our X-ray crystal structures suggest a potential mechanism for dimyristoylation. We observed that the acetyl of the N-terminus of lysine myristoylated peptide points into a hydrophobic region formed by the rings of the three tyrosine residues (Supplementary Fig. 6B). Following that region, we found a long hydrophobic pocket (~23 Å) that emerges from the active site of NMT1 and

NMT2 (Fig. 2f, Supplementary Fig. 6C). In both enzymes, this pocket is surrounded by two phenylalanines, four leucines, one valine, one alanine, one methionine, one asparagine, six tyrosines, and one isoleucine. From the Lys3 $C_\alpha$, the pocket is approximately the same length as the myristoyl-lysine pocket. In our structure, it is occupied by glycerol and nine waters, but it could potentially fit a myristoylated glycine (as modeled in Fig. 2g), while orienting Lys3 for a second myristoylation reaction. This pocket is present in other NMT structures such as 5O9V but was previously unnoticed. While similar pockets in other GNAT family of enzymes are hypothesized to be water channels that function in deprotonation of the nucleophile[29], we speculate that in NMT it might also facilitate the second myristoylation event. Furthermore, NMT uniquely contains two GNAT domains thought to result from gene duplication[30], the hydrophobic pocket that we found is located in that domain, and so it is tempting to propose that the second domain, while has lost its catalytic activity, is retained to hold a myristoyl moiety.

**NMT might have other lysine myristoylation substrates**. We then explored NMT sequence requirement for lysine myristoylation. We first inserted one glycine or alanine or two alanine residues between A2 and K3 of the ARF6 G2A mutant and performed Alk12 labeling in HEK293T cells overexpressing NMT2 or with NMT inhibition. While the insertion of one G was tolerated, the insertion of an alanine significantly decreased myristoylation by NMT (Fig. 3a). A mutant containing two additional alanine residues (Fig. 3a, lanes 6, 10, 14) could not be stably expressed. In vitro reactions on ARF6 N-terminal peptides with similar sequence changes led to the same conclusion (Fig. 3b). This suggests that NMTs may regulate other substrates with lysine at position 3 or 4. Interestingly, in cells we also identified a peptide with A2 acetylated and K3 myristoylated which, based on the MS/MS quantification, was about four times more abundant than the K3 myristoylated peptide without the acetylation (Supplementary Fig. 5). This suggests that NMT can accommodate substrates with other N-terminal acylations in vivo.

We then varied amino acids within the ARF6 N-terminus on synthetic peptides and tested whether they remain NMT1 and NMT2 substrates. This revealed a strong preference for S6 and K7, which has been reported for N-terminal glycine myristoylation, and a lack of tolerance for charged residues at position 5 (Fig. 3c). This is supported by our NMT2 structure where the S6 side chain of the substrate is specified by His296 side chain and the Gly472 backbone nitrogen, the K7 side chain is specified by a pocket of aspartates (183, 185, and 471), and residue 5 experiences favorable hydrophobic interactions with nearby phenylalanine residues 188, 190, and 311 as well as Leu416. Main chain contacts between NMT and peptide substrate residues 7–9 are also conserved (Fig. 3e). Interestingly, when the sequence was changed to mimic the ARF1 sequence but with lysine in position 3 (sequence AKIFANLFWW), the reaction did not occur on a peptide in vitro (Fig. 3c) or on the protein in cells (Fig. 3d). Only when S6 and K7 were introduced (sequence AKIFSKLFWW) did the reaction happen in vitro (Fig. 3c, last entry). These observations suggest strong similarities in NMT N-terminal glycine and lysine substrate recognition mechanisms with more restrictions for lysine myristoylation.

**SIRT2 removes ARF6 K3 myristoylation**. In order to further study the biological significance of ARF6 K3 myristoylation, we sought to identify the eraser of this modification. This is important as NMT manipulation will affect both glycine and lysine myristoylation and thus make it difficult to analyze the contribution of lysine myristoylation. Lysine fatty acylation is

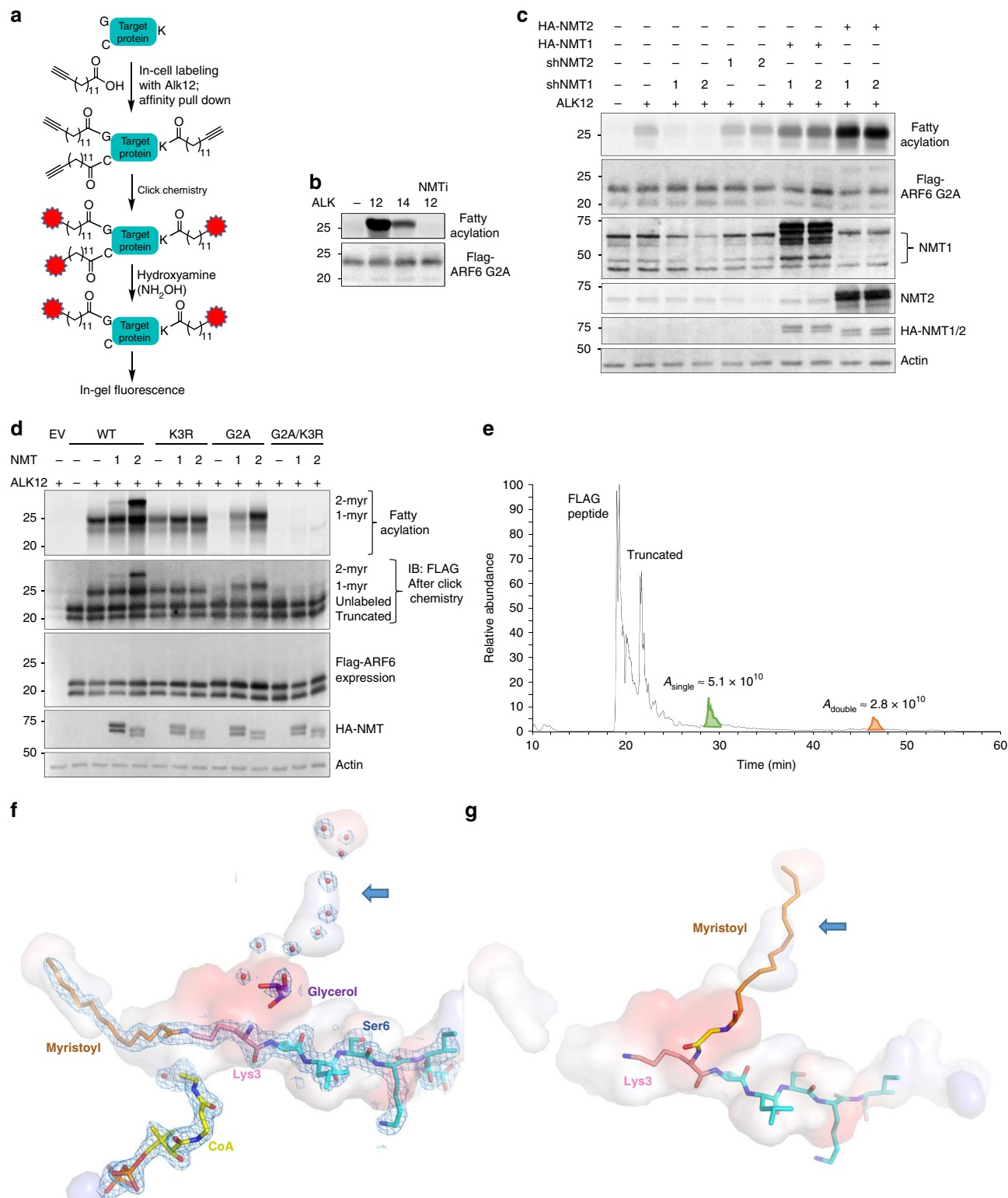

**Fig. 2 Biochemical and structural evidence for the NMT activity on lysine. a** In-cell Alkyne probe labeling scheme. **b** Alk12 labeling is more efficient than Alk14 labeling and NMT inhibition abolishes labeling. **c** NMT1 but not NMT2 knockdown inhibits lysine myristoylation in HEK293T cells. **d** NMT1 and NMT2 overexpression produce singly and doubly myristoylated ARF6 based on Alk12 labeling. **e** Total ion chromatogram from a top–down mass spectrometric analysis that reveals di-myristoylated ARF6 isolated from HEK293T cells with NMT2 overexpression. Confirmatory MS/MS data are found in Supplementary Fig. 4. **f** Interior pockets (shown as surface) around the NMT2 active site, showing possible channel for second myristoyl (blue arrow) currently occupied by waters (red dots) and glycerol (purple). The surface is colored by the electrostatic properties of the surrounding residues: blue (positive), red (negative), and gray (hydrophobic). The 1.3 σ $2F_O$–$F_C$ electron density map is shown as a mesh. **g** Model of myristoyl-glycine docked in the second pocket, with lysine ready for a second myristoylation reaction.

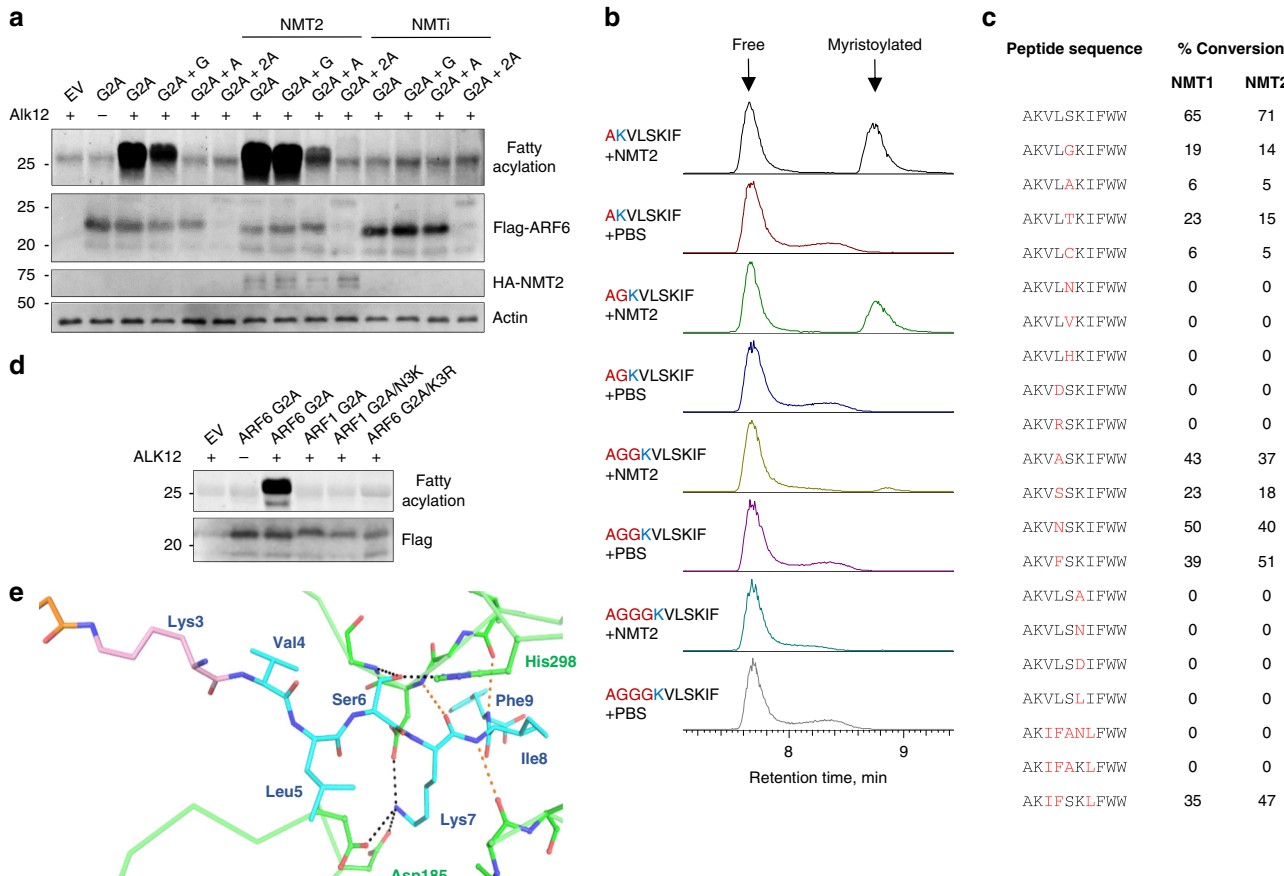

**Fig. 3 Sequence requirement for NMT lysine myristoylation. a** Cellular Alk12 labeling results with the indicated mutants showing that NMT can accommodate other positions of K at the N-terminus in cells. The mutant containing two additional alanine residues at the N-terminus could not be stably expressed. **b** NMT can accommodate other N-terminal positions of K in vitro on synthetic peptides. Ion chromatograms searched for the substrate and product ions are shown. **c** Amino acid sequence preference for NMT lysine myristoylation on ARF6 G2A-derived peptides. Red letters represent changed residues. The sequence IFANL is the ARF1-derived sequence. The product was detected by LC-MS and the percentage of conversion was calculated from the substrate and product peak areas on HPLC UV traces. **d** Alk12 labeling of the indicated mutants transiently overexpressed in HEK293T cells showing that, unlike ARF6 G2A, ARF1 G2A/N3K is not myristoylated. **e** Peptide recognition for Lys3 myristoylation appears similar to that for Gly2 myristoylation. NMT2 residues are shown in green. Hydrogen bonding to Ser6 by His298 and Gly472 backbone nitrogen and to Lys7 side chain by a cluster of aspartates 183, 185, and 471 are shown as black dotted lines. Further hydrogen bonding to the backbone of the peptide (orange dotted lines) is also the same as in previous structures.

reversible and several sirtuins were identified as lysine defatty acylases[1,2,4,5]. Given that SIRT2 is the only primarily cytosolic sirtuin, we asked whether SIRT2 could be the eraser of ARF6 lysine myristoylation. SIRT2 OE removed ARF6 lysine myristoylation in cells overexpressing NMT (Fig. 4a). To confirm the direct deacylation by SIRT2, we purified ARF6 WT and G2A from cells overexpressing NMT2 and treated with Alk12 and then treated the purified ARF6 proteins with recombinant SIRT2 in the presence of NAD[+]. This treatment strongly decreased lysine myristoylation (Fig. 4b). We then asked whether other known demyristoylases can remove ARF6 lysine modification in HEK293T cells. We performed Alk12 labeling of ARF6 G2A in SIRT1, SIRT3, SIRT6, SIRT7, and HDAC11 stable KD HEK293T cells using SIRT2 KD and TM, nicotinamide (NAM), or SAHA treatments as controls. Only SIRT2 KD, TM, and NAM increased ARF6 G2A labeling suggesting that SIRT2 is the major eraser of ARF6 lysine myristoylation in HEK293T cells (Supplementary Fig. 14).

To further confirm that lysine myristoylation of ARF6 is not an artifact of NMT OE, we used a previously reported sensitive [32]P-NAD[+] assay to detect ARF6 myristoylation without overexpressing NMT[31]. We isolated ARF6 WT and K3R mutants from

SIRT2 KD HEK293T cells and treated them with recombinant SIRT2 in the presence of [32]P-NAD[+] (Fig. 4c). Myristoyl-H3K9 peptide, a known in vitro substrate of SIRT2, was used as a positive control. Separation of the reaction products by thin-layer chromatography (TLC) revealed a myristoyl ADP-ribose product (My-ADPR) in the reaction containing ARF6 WT but not K3R mutant (Fig. 4d). Furthermore, SIRT2 in the presence of NAD[+] could remove K3 myristoylation but not G2 myristoylation from synthetic peptides (Supplementary Fig. 8A, B). This strongly supports that ARF6 WT is myristoylated on K3 by endogenous NMT. We also used the [32]P-NAD[+] assay to confirm that SIRT2 inhibition with a SIRT2-specific inhibitor TM[32] in cells (Fig. 4e) and NMT OE or in vitro NMT treatment (Supplementary Fig. 7A, B) can increase ARF6 K3 myristoylation. In addition, KD and inhibition of SIRT2 with TM increased the levels of ARF6 G2A lysine myristoylation, and this effect was rescued by SIRT2 OE (Fig. 4f). Finally, co-immunoprecipitation (co-IP) studies suggested that ARF6 and SIRT2 interact (Supplementary Fig. 8C). These data demonstrate that SIRT2 is the eraser of ARF6 K3 myristoylation.

With this knowledge, we sought to confirm that one of the two single myristoylation bands generated by the NMT on ARF6 WT

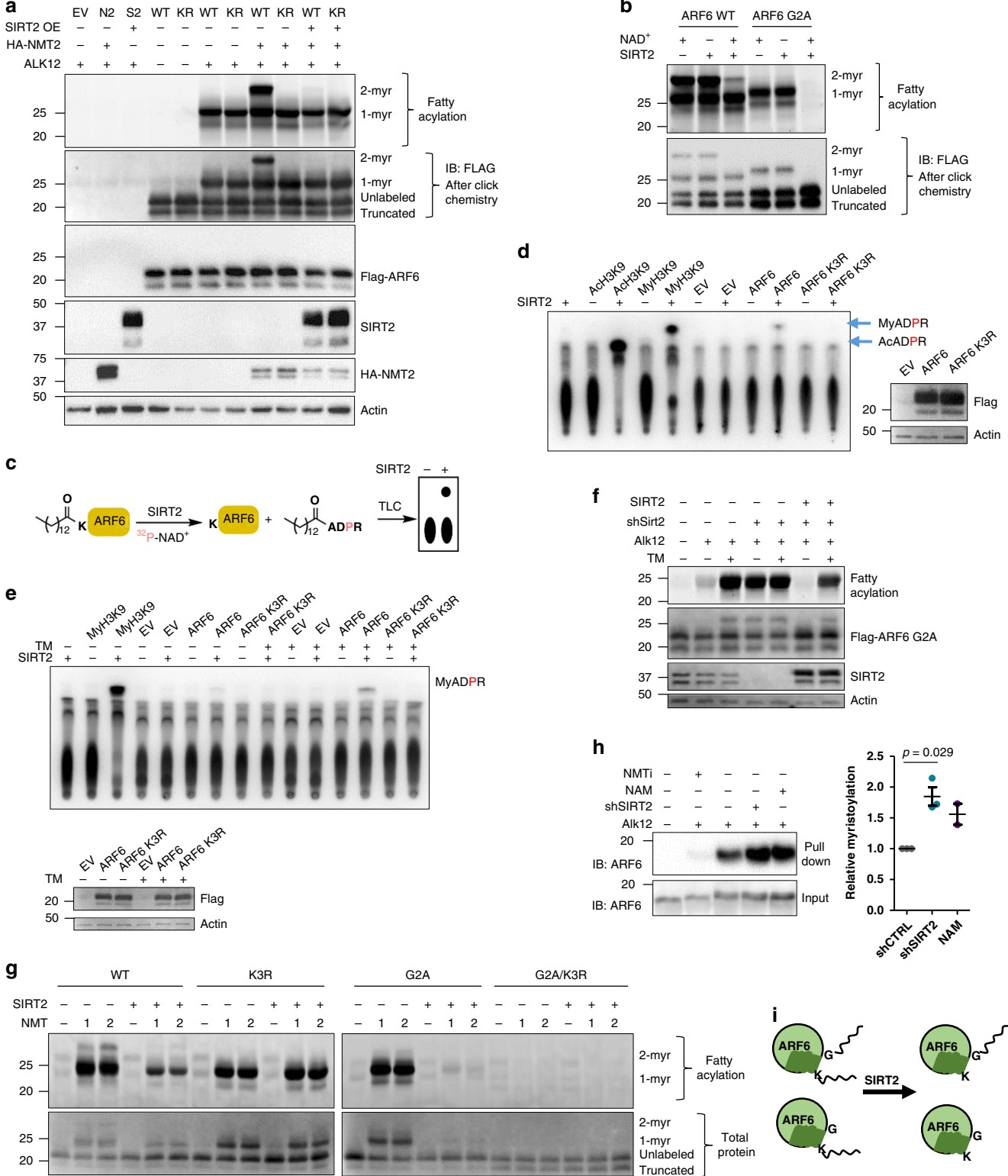

in vitro (Fig. 1e) is lysine myristoylated ARF6. To achieve that, we reconstituted the NMT reaction on ARF6 WT and mutants with Alk12-CoA and followed with a SIRT2 reaction. SIRT2 removed the double acylation and the top half of single acylation bands from ARF6 WT and most of the modification from ARF6 G2A leaving the K3R and G2A/K3R mutants unaffected (Fig. 4g). This confirms that NMT myristoylation on K3 of ARF6 protein may not require N-terminus sequestration and can occur to an extent similar to glycine myristoylation.

Next, we tested whether endogenous ARF6 is myristoylated on K3. Since we were unable to efficiently isolate endogenous ARF6 with commercial antibodies, we labeled ARF6 with Alk12 in cells with depleted or inhibited SIRT2. We then removed cysteine labeling in lysates with hydroxylamine, conjugated biotin azide followed by streptavidin pull down. Western blot (WB) analysis revealed a signal increase with SIRT2 KD or sirtuin inhibitor NAM (Fig. 4h), suggesting a higher abundance of lysine-modified species. Together these data suggest that endogenous ARF6

**Fig. 4 SIRT2 removes ARF6 K3 myristoylation. a** Alk12 labeling results on overexpressed ARF6 WT or K3R mutant (KR) showing that SIRT2 demyristoylates ARF6 in HEK293T cells as indicated by the disappearance of the dimyristoylated species with SIRT2 OE. **b** SIRT2 demyristoylates ARF6 and ARF6 G2A in vitro. ARF6 WT and G2A were isolated from Alk12 treated cells with NMT2 overexpression and were treated with recombinant SIRT2 followed by TAMRA azide conjugation and in-gel fluorescence. **c** $^{32}$P-NAD$^+$ assay scheme. **d** SIRT2 demyristoylates ARF6 K3 purified from cells with endogenous NMT in vitro (detected using the $^{32}$P-NAD$^+$ assay). **e** SIRT2 inhibition with TM in cells increases lysine fatty acylation of ARF6 (detected using the $^{32}$P-NAD$^+$ assay on ARF6 isolated from HEK293T cells treated with TM). **f** Alk12 labeling in HEK293T cells transiently overexpressing ARF6 G2A showing that SIRT2 KD or inhibition increase but SIRT2 OE removes the labeling. **g** SIRT2 removes double and single lysine myristoylation produced by in vitro acylation with NMT1/2. The indicated ARF6 proteins were isolated from cells with NMT inhibition and were first modified by NMT with Alk12-CoA in vitro and then reacted with SIRT2 in vitro. **h** Endogenous ARF6 has lysine myristoylation in HEK293T cells. Fifteen-hour treatment with 10 mM nicotinamide or 2 μM DDD85646 was used. Endogenous myristoylated proteins were labeled with Alk12 followed by conjugation to biotin azide via click chemistry and streptavidin pull down. The pull down products were analyzed by western blot for ARF6. Data points represent biological replicates analyzed by unpaired two-tailed *t* test. Error bars represent SEM. **i** Model showing that SIRT2 can remove ARF6 K3 myristoylation.

contains lysine myristoylation and SIRT2 is its physiological eraser (Fig. 4i).

**NMT prefers ARF6-GTP while SIRT2 prefers ARF6-GDP.** Because ARF6 cycles between GTP- and GDP-bound states, we reasoned that lysine myristoylation might need to be removed at a specific point to support the GTPase cycle. We therefore examined the ability of SIRT2 to act on active Q67L and inactive T27N mutants of ARF6 isolated from SIRT2 KD cells via $^{32}$P-NAD$^+$ assay. More My-ADPR was formed in the reaction with the T27N mutants suggesting that SIRT2 might have a preference for the GDP-bound or nucleotide-free ARF6 (Fig. 5a). However, this could also indicate that ARF6 T27N contains more lysine myristoylation. To address that, we examined the abundance of lysine myristoylation by measuring the relative ratio of double to single myristoylation fluorescent bands of ARF6 Q67L and T27N. In control cells overexpressing NMT2, T27N had much less dimyristoylation compared to that of Q67L (Fig. 5b and Supplementary Fig. 12). T27N dimyristoylation strongly increased in SIRT2 KD cells, yet was less abundant than that on Q67L. Unlike Q67L dimyristoylation, T27N dimyrisotylation was completely removed with SIRT2 OE (Fig. 5b). Since the abundance of lysine myristoylation on T27N was not higher than that on Q67L (Fig. 5b) but more My-ADPR was formed during SIRT2 in vitro reaction with T27N (Fig. 5a), the data suggest that SIRT2 prefers the inactive GDP-bound ARF6.

We next asked whether the addition of this modification is also GTPase cycle dependent. In vitro using purified proteins, the reactions of NMT with the active ARF6 Q67L mutant produced more doubly myristoylated ARF6 (Fig. 5c). In cells, co-IP experiments showed that NMT2 preferentially binds to Q67L over T27N (Fig. 5d) and NMT2 co-localizes with ARF6 Q67L better than with ARF6 T27N (Fig. 5e). Together, these results suggest that NMT prefers the GTP-bound ARF6. Even without NMT OE, we were able to detect about 3% of dimyristoylated ARF6 Q67L (Supplementary Fig. 16). We anticipate that endogenous ARF6-GTP may have a higher relative dimyristoylation due to a higher enzyme/substrate ratio; however, we could not estimate this value due to technical limitations. Overall, our data support that lysine myristoylation is added and removed at distinct points of the ARF6 GTPase cycle (Fig. 5f). Furthermore, Fig. 1 suggests that on a synthetic ARF6 peptide lysine myristoylation by NMT is relatively inefficient compared to that on ARF6 protein in vitro or in cells (Figs. 2d, e and 4g). The preference of NMT for ARF6-GTP could potentially explain these observations.

**K3 myristoylation regulates ARF6 localization.** The ARF GTPase membrane binding is regulated by a glycine myristoyl switch where upon GTP binding the myristoylated amphipathic N-terminal helix unfolds and inserts into the membrane bilayer. Two fatty acyl chains are often required for membrane binding of a number of proteins such as those in the Src and G alpha families[33,34]. We asked whether ARF6 requires a second fatty acyl group for efficient membrane binding. To this end, we first compared the localization of ARF6 WT and K3R in cells with NMT2 OE by immunofluorescence (IF). ARF6 WT localized more to the cell periphery compared to ARF6 K3R (Fig. 6a). We then compared the abundance of ARF6 WT and K3R in the membrane and cytosolic fractions in HEK293T cells with NMT2 OE. The K3R mutation decreased the abundance of ARF6 in the membrane fraction but increased its levels in the cytosol, and the doubly acylated ARF6 was mostly found in the membrane fraction suggesting that K3 myristoylation targets ARF6 to membranes (Fig. 6b and Supplementary Fig. 13A).

ARFs 1–5 are cytosolic when GDP bound; however, ARF6 largely remains on the membrane when inactivated[24,35,36], which has been a longstanding puzzle. We therefore hypothesized that lysine myristoylation contributes to the membrane association of inactive ARF6. It has been shown that the GAP-catalyzed GTP hydrolysis of ARF6 is necessary for its transport to the endocytic recycling compartment (ERC) via the endocytic pathway, while the GDP to GTP exchange catalyzed by GEFs is necessary for its recycling to the plasma membrane[37]. Subcellular fractionation of ARF6 WT, Q67L, and T27N in SIRT2 KD and NMT2 OE cells revealed that the Lys3 myristoylated ARF6 proteins were present in the membrane fraction more than the corresponding K3R mutants that cannot be myrisotylated (Fig. 6c and Supplementary Fig. 13B). Furthermore, in cells without NMT OE ARF6 T27N K3R was less abundant on the membrane than ARF6 T27N supporting that endogenous NMT and SIRT2 regulate the membrane binding cycle through K3 myristoylation of inactive ARF6 (Supplementary Fig. 13C).

We then examined the localization of Q67L and T27N to the plasma membrane and the ERC located at the perinuclear region, where they are known to reside, by microscopy[37,38]. We used TfR, a marker for plasma membrane and ERC[39,40], to examine the colocalization of ARF6 mutants. Q67L did not localize to ERC but was found at the plasma membrane and plasma membrane folds under it and blocked TfR trafficking as was observed by others (Fig. 6d)[37,41,42]. Changing K3 to R3 in the Q67L mutant had very little effect on its colocalization with TfR, indicating that N-terminal glycine myristoylation might be sufficient for the correct cellular localization of GTP-bound ARF6. The T27N mutant is localized to the plasma membrane as well as the early endosomes and perinuclear ERC, as indicated by its colocalization with TfR[37,38]. However, the T27N K3R mutant, unlike T27N, was mostly cytosolic with little localization to ERC, suggesting that lysine myristoylation is necessary for its trafficking to this compartment. Consistent with this, SIRT2 KD increased, while

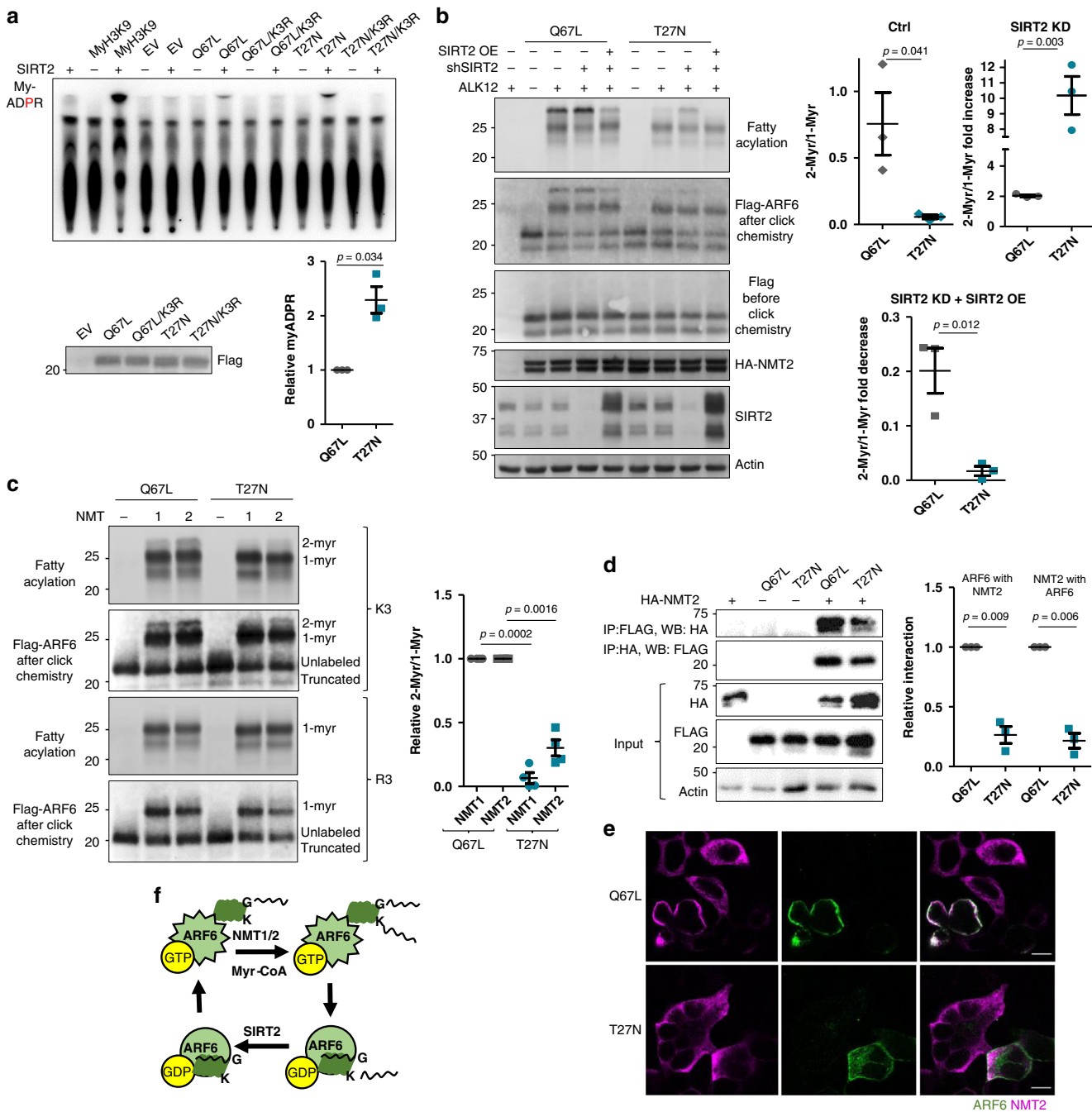

**Fig. 5 Lysine myristoylation cycle is coupled to the catalytic cycle of ARF6. a** SIRT2 removes more lysine myristoylation from ARF6 T27N than from ARF6 Q67L in vitro. ARF6 proteins were purified from cells with endogenous NMT and SIRT2 KD and subjected to the $^{32}$P-NAD$^+$ assay with recombinant SIRT2. $n = 3$. **b** In-cell Alk12 labeling of ARF6 Q67L and T27N mutants during NMT2 OE showing more lysine acylation of ARF6 Q67L. $n = 3$. **c** In vitro NMT reaction on ARF6 Q67L and T27N showing more double acylation on ARF6 Q67L (ARF6-GTP). $n = 4$. **d** NMT2 preferentially binds ARF6-GTP. Flag-tagged ARF6 mutants and HA-NMT2 were overexpressed in HEK293T cells and subjected to the indicated pull downs. $n = 3$. **e** NMT2 is redistributed to the sites of ARF6 Q67L, but not ARF6 T27N. Confocal images of immunofluorescence in cells overexpressing FLAG-ARF6 mutants and HA-NMT2. Scale bars: 10 μm. **f** Model showing that NMT acts on active, but SIRT2 on inactive ARF6. For all experiments, error bars represent SEM and $n$ represents biological replicates analyzed by unpaired two-tailed $t$ test.

NMT inhibition or NMT1 KD (but not NMT2 KD) decreased, ARF6 T27N colocalization with TfR (Supplementary Fig. 10 and 11). Interestingly, we observed SIRT2 colocalization with TfR in HEK293T cells under basal conditions (Fig. 6e), which suggests that SIRT2 might act on ARF6 at ERC or early endosomes where TfR is known to be present[43]. Since inactive ARF6 localizes to ERC, this further supports that SIRT2 regulates inactive ARF6. NMT2 appeared largely cytosolic or on endomembranes (Fig. 6a)

suggesting that it might act on ARF6 before it is trafficked to the plasma membrane.

**Lysine myristoylation cycle promotes ARF6 activity.** Since our data suggest that lysine myristoylation increases ARF6 membrane localization, we asked whether this also promotes ARF6 GTP loading by increasing ARF6 interaction with GEFs at the membranes. To test this, we performed an ARF6-GTP pull down with

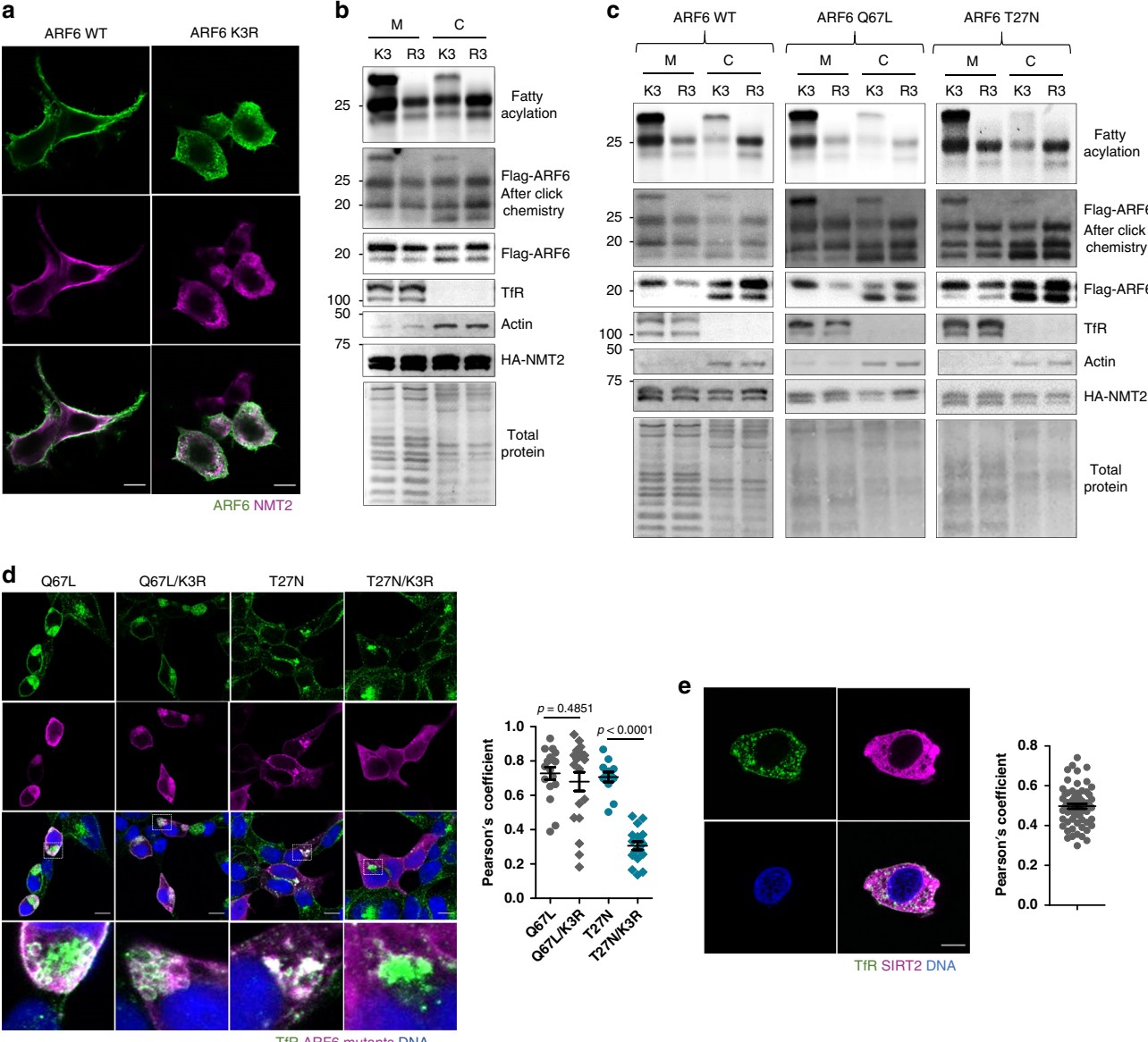

**Fig. 6 Lysine myristoylation regulates ARF6 membrane localization. a** Confocal microscopy of overexpressed Flag-ARF6 and HA-NMT2 showing a decreased membrane localization of ARF6 K3R compared to ARF6 WT and colocalization of ARF6 with NMT2 at the plasma membrane and cytosol but not ERC in HEK293T cells. Scale bars: 10 μm. **b** Subcellular fractionation results in cells with HA-NMT2 expression and Alk12 treatment showing that K3 myristoylation promotes ARF6 membrane localization. The K3R mutant shows a decreased membrane and increased cytosolic localization compared to ARF6 WT. $n = 2$. **c** Subcellular fractionation results in cells with HA-NMT2 expression, SIRT2 KD, and Alk12 treatment demonstrating that lysine myristoylation promotes ARF6 membrane localization in different catalytic states. $n = 2$. **d** ARF6 K3R mutation does not affect ARF6 Q67L localization but inhibits ARF6 T27N localization to plasma membrane and ERC as indicated by colocalization with TfR in SIRT2 KD cells. These cells do not overexpress NMT and thus we were examining the effect of endogenous lysine myristoylation. Each point represents one cell (Q67L = 17, Q67L/K3R = 19, T27N = 12, T27N/K3R = 18). Scale bars: 10 μm. **e** SIRT2 localizes to ERC, recycling, and early endosomes as indicated by colocalization with TfR. Each point represents one cell (63 cells). Scale bars: 10 μm. All experiments in this Fig. were performed in HEK293T cells. Colocalization quantifications are presented as Pearson's correlation coefficients. For all experiments, error bars represent SEM and $n$ represents biological replicates analyzed by unpaired two-tailed $t$ test.

GGA3, a known ARF6 effector protein, conjugated agarose and examined its levels by WB. NMT inhibition suppressed but OE promoted ARF6 activation (Fig. 7a). To determine the contribution of lysine myristoylation, we examined the effects of SIRT2 inhibition and K3R mutation. SIRT2 inhibition reduced GTP loading of overexpressed ARF6 WT but not K3R (Fig. 7b), suggesting that SIRT2 demyristoylates ARF6 K3 to promote ARF6 activation.

To rule out that lysine myristoylation or K3R mutation itself affect binding to GGA3, we examined the structures of

ARF1 in complex with the GGA1-binding domain as ARF6-GGA3 structures are not available. This revealed that the N-terminal helix is unlikely to participate in this interaction (Supplementary Fig. 9A). Furthermore, it is reported that, like other effector proteins, GGA3 binds the interswitch region that extrudes upon GTP binding along with switch 1 and 2 but not N-terminal helix[44]. Finally, we observed that SIRT2 inhibition did not affect the binding of the active ARF6 Q67L mutant to GGA3 (Supplementary Fig. 9B), which further

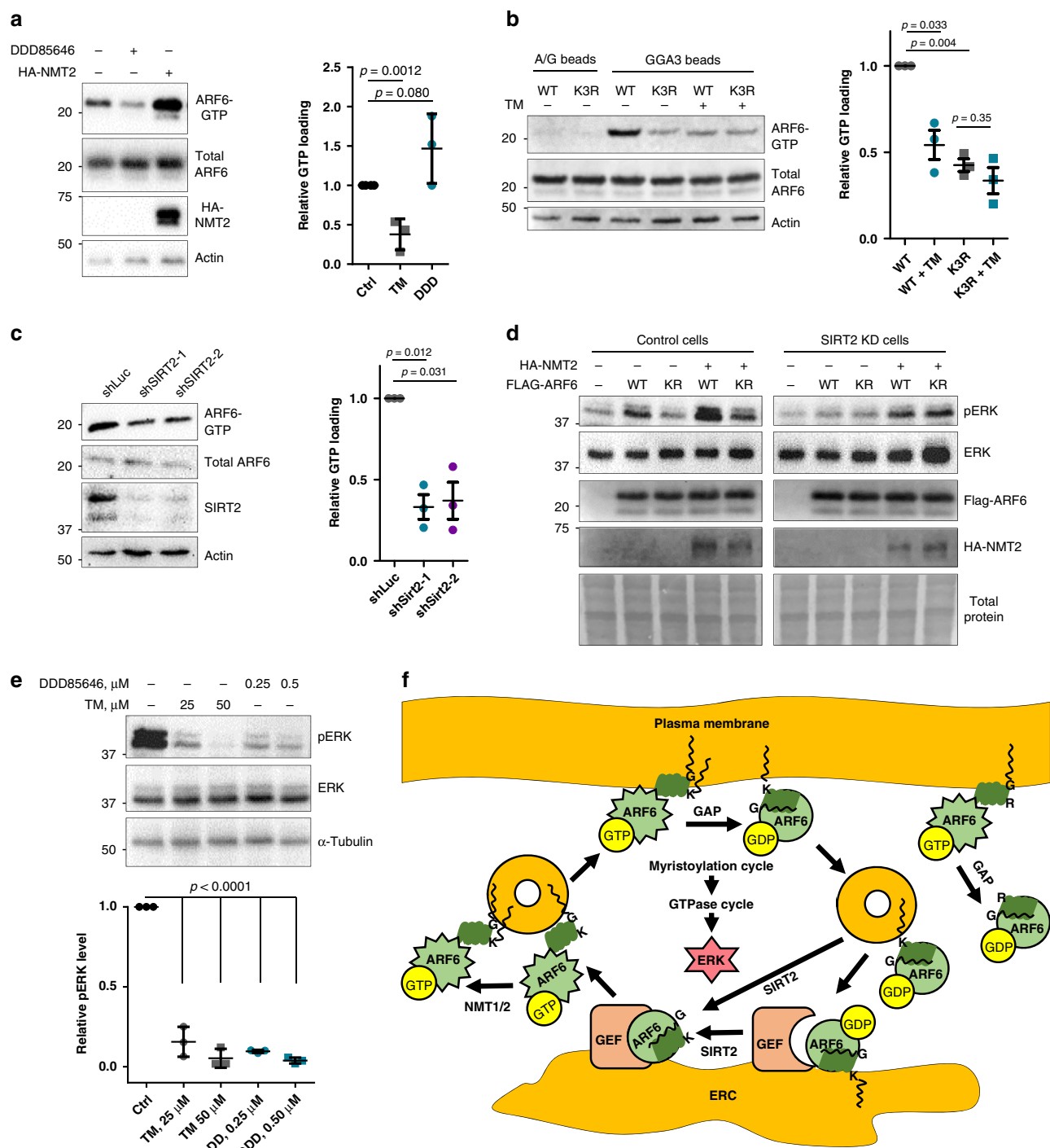

**Fig. 7 Lysine myristoylation cycle regulates ARF6 activation to control ERK phosphorylation. a** NMT inhibition with DDD85646 suppresses, but NMT2 OE promotes, ARF6 GTP loading. GGA3 pull down in cells expressing Flag-ARF6 and subjected to NMT2 OE or NMTi. $n = 3$. **b** GGA3 pull down showing that SIRT2 inhibition by TM decreases the GTP loading of ARF6 WT, but not K3R. $n = 3$. **c** SIRT2 KD inhibits GTP loading of endogenous ARF6. GGA3 pull down was performed in HEK293T cells with transient SIRT2 KD. $n = 3$. **d** ARF6 lysine myristoylation promotes pERK in control but not in SIRT2 KD HEK293T cells. ARF6 WT or K3R mutant were overexpressed in HEK293T cells that were blotted for pERK after 15 h of serum starvation. $n = 2$ **e** SIRT2 and NMT inhibition dose-dependently inhibit pERK. HEK293T cells were treated with the indicated inhibitors for 15 h. $n = 3$. For all experiments, error bars represent SEM and $n$ represents biological replicates analyzed by unpaired two-tailed $t$ test. **f** Model for the coupling of ARF6 myristoylation–demyristoylation cycle and GTPase cycle. NMT myristoylates ARF6-GTP on K3, which targets ARF6 to plasma membrane and retains inactive ARF6 at the membrane after GTP hydrolysis allowing its trafficking to ERC via the endocytic pathway. SIRT2 deacylates inactive ARF6 at early endosomes or ERC to allow its efficient activation by GEFs after GDP release. ARF6-GTP on recycling endosomes gets myristoylated on K3 by NMT, which drives its plasma membrane translocation. This cycle in turn controls ERK phosphorylation. ARF6 K3R cannot be myristoylated on K3 and therefore loses its membrane association after GTP hydrolysis, which inhibits its translocation to endomembranes and activation.

confirms that lysine myristoylation itself does not interfere with GGA3 binding.

To find out whether the activity of endogenous ARF6 is regulated by lysine myristoylation, we examined the effect of SIRT2 KD on GTP loading of native cellular ARF6. SIRT2 KD suppressed the abundance of GTP-bound endogenous ARF6 suggesting that this regulation is not an artifact of OE (Fig. 7c).

We were initially surprised that ARF6 WT (which can be K3 myristoylated) contained more ARF6-GTP than the K3R mutant (which cannot be K3 myristoylated) in cells with active SIRT2, but the GTP loading for ARF6 WT was diminished when SIRT2 was inhibited and thus K3 myristoylation accumulation (Fig. 7b). However, given the observed preference of NMT for ARF6-GTP and of SIRT2 for ARF6-GDP, this unexpected GTP loading result suggests that the myristoylation–demyristoylation cycle serves to drive the ARF6 GTPase cycle. Thus disrupting either myristoylation or demyristoylation inhibits ARF6 activation (Fig. 7f).

The GGA3–ARF6 interaction is known to promote extracellular signal-regulated kinase (ERK) activation[45]. Since the GGA3 pull down assay demonstrated that this interaction is regulated by lysine myristoylation, we examined the effect of ARF6 K3 myristoylation on ERK phosphorylation in HEK293T cells. Serum-starved cells were used to isolate the lysine myristoylation effect on ARF6 activation from other activating signals such as growth factors. Consistent with the GTP-loading data, ARF6 WT OE promoted pERK that was further increased by NMT2 OE in control cells, while K3R had little effect (Fig. 7d). Similarly, SIRT2 KD decreased pERK for ARF6 WT but not K3R (Fig. 7d). Furthermore, both NMT and SIRT2 inhibition in HEK293T cells inhibited ERK phosphorylation suggesting that myristoylation cycle of endogenous ARF6 may regulate this pathway (Fig. 7e).

Together these data suggest a model (Fig. 7f) where NMT promotes ARF6 plasma membrane localization by myristoylating ARF6-GTP on K3, and after GAP-catalyzed GTP hydrolysis the modification keeps the inactive ARF6 membrane bound. This allows its trafficking to intracellular vesicles including the ERC, where lysine demyristoylation by SIRT2 supports the efficient activation of ARF6 by GEF and subsequent trafficking back to plasma membrane. ARF6 K3R fails to retain membrane association after GTP hydrolysis and therefore cannot be efficiently trafficked and activated. This model readily explains the GTP-loading data in Fig. 7 and the importance of active myristoylation–demyristoylation cycle for the normal GTPase cycle of ARF6 to control downstream signaling.

## Discussion

Our work uncovered an ARF6 lysine myristoylation–demyristoylation cycle, which is intimately coupled to its GTPase cycle and controlled by SIRT2, and a previously unknown activity of NMT. NMT enzymes myristoylate ARF6 on K3 to promote its membrane association and trafficking to ERC while SIRT2 removes this moiety to allow efficient ARF6 activation. This in turn regulates ARF6 activity-dependent effector binding and downstream signaling. GEFs of ARF6 are known to localize to the plasma membrane and ERC[46,47]. The increased membrane localization of ARF6 by K3 myristoylation ensures the endocytic trafficking of ARF6 to increase its encounters with its GEFs at endomembranes, but for efficient GTP loading, K3 myristoylation needs to be removed by SIRT2. The selectivity of NMT for ARF6-GTP and SIRT2 for inactive ARF6 accelerates ARF6 activation by avoiding a futile lysine myristoylation–demyristoylation cycle (Fig. 7f). While our study provides a mechanism for ARF6 unique plasma

membrane targeting, lysine myristoylation may not be its sole driver, since ARF6 K3R did not localize to the Golgi, like other ARFs. Membrane receptors or GAP and GEF specificities might aid this targeting. Since, unlike glycine modification, lysine myristoylation is dynamic, its lower abundance compared to glycine myristoylation is not surprising. Although we were unable to resolve dimyristoylated endogenous ARF6 with currently available methods, we observed that SIRT2 KD or inhibition increases ARF6 myristoylation and regulates its GTP loading and downstream signaling.

ARF6–pERK axis controls an array of cellular activities, such as migration[48], tubule development[49], and vesicle shedding[50]. Given our findings, these processes can be modulated by targeting the ARF6 myristoylation–demyristoylation cycle with selective inhibitors for NMT and SIRT2, offering new therapeutic strategies.

The lysine myristoyltransferase activity of NMTs tolerates small changes in the substrate sequence supporting the existence of other lysine substrates. Furthermore, conditions regulated by proteolysis such as apoptosis, immune response, and viral infections might generate additional NMT lysine substrates. Our findings will facilitate the discovery of other proteins regulated by lysine fatty acylation and thus open new avenues for understanding the biological functions of this modification.

## Methods

**Reagents**. Anti-FLAG affinity gel (#A2220, RRID: AB_10063035) and FLAG-HRP (#A8592, RRID: AB_439702, 1:5000 dilution) were purchased from Sigma. HA-HRP (F-7, sc-7392, 1:5000 dilution), Na/K-ATPase (C464.6, sc-21712, 1:1000 dilution), β-Actin-HRP (C4, sc-47778, 1:5000 dilution), NMT1 (E-9, sc-393702, 1:1000 dilution), NMT2 (30, sc-136005, 1:1000 dilution), TfR (CD-71) (3B8 2A1, sc-32272, 1:500 dilution for WB and 1:200 dilution for IF), and Arf6 (3A-1, sc-7971, 1:1000 dilution) antibodies were purchased from Santa Cruz Biotechnology and SIRT2 (D4050, 12650S, 1:1000 dilution), SIRT6 (D8D12, 12486S, 1:1000 dilution), SIRT7 (D3K5A, 5360S, 1:1000 dilution), SIRT1 (D739, 2493S, 1:1000 dilution), SIRT3 (D22A3, 5490S, 1:1000 dilution), Arf6 (D12G6, 5740, 1:1000 dilution), Phospho-p44/42 MAPK (Erk1/2) (Thr202/Tyr204) (D13.14.4E, 4370, 1:1000 dilution), and p44/42 MAPK (Erk1/2) (L34F12, 4696, 1:2000 dilution) antibodies from Cell Signaling Technology.

$^{32}$P-NAD$^+$ was purchased from PerkinElmer, Tris[(1-benzyl-1H-1,2,3-triazol-4-yl)methyl]amine (TBTA), Tris(2-carboxyethyl)phosphine (TCEP), hydroxylamine, NAD$^+$, and protease inhibitor cocktail were purchased from Sigma. FuGene 6 transfection reagent were purchased from Promega (Madison, WI). ECL plus WB detection reagent were purchased from Thermo Scientific Pierce (Rockford, IL). 5-Tamra azide was purchased from Lumiprobe (47130) and GGA3-PBD beads from cytoskeleton (GGA05-A). Polyethylenimine (PEI) MAX transfection reagent (24765-1) was purchased from Polysciences. NMT1, NMT2, and SIRT2 lentiviral plasmids (pLKO.1-puro vector) were purchased from Sigma, the sequences of short hairpin RNA (shRNA) are: NMT1-1 TRCN0000035713 (CCGGCCTGAGCAGAAATATGACCATCTCGAGATGGTCATATT TCTGCTCAGGTTTTTG); NMT1-2 TRCN0000289868 (CCGGCGGAAATTGG TTGGGTTCATTCTCGAGAATGAACCCAACCAATTTCCGTTTTTG); NMT2-1 TRCN0000291915 (CCGGCCAACGGTAAACTGACTGATTCTCGAGAATCA GTCAGTTTACCGT TGGTTTTTG); NMT2-2 TRCN0000303312 (CCGGGAAA TTGAAGTAGTCGATAATCTCGAGATTATCGAC TACTTCAATTTCTTTT TG); SIRT2-1 TRCN0000040219 (CCGGGCCATCTTTGAGATC AGCTATC TCGAGATAGCTGATCTCAAAGATGGCTTTTTG), SIRT2-2 TRCN0000310335 (CCGGCCTGTGGCTAAGTAAACCATACTCGAGTATGGTTTACTTAGCCA CAGGTTTTTG).

Alk12 and Alk14 were synthesized as reported[28]. TM was synthesized as previously reported[32]. Synthetic peptides: GKVLSKIF, AKVLSKIF, AKVLSKIFWW, AK(myr)VLSKIFWW, G(myr)KVLSKIFWW, and GK(myr) VLSKIFWW and the peptides listed in Fig. 3c were purchased from Biomatik; acetyl-GKVLSKIF, acetyl-KVLSKIF, KVLSKIF, RKVLSKIF, AGKVLSKIF, AGGKVLSKIF, and AGGGKVLSKIF were synthesized in-house using a peptide synthesizer.

**Cell culture and transient transfection**. Human HEK293T cells (obtained from ATCC) were cultured in Dulbecco's modified Eagle's medium with 10% heat-inactivated fetal bovine serum (10437028, Thermo Fisher) or calf serum (18439-24-2, Sigma). Transient transfection was done using FuGene 6 or PEI. Briefly, 1:3 ratio of plasmid (μg) to transfection reagent (μl) was used. Transfection reagent was added to serum-free media (10% volume of culture media) followed by a 5-min incubation at room temperature for FuGene 6. Then the plasmid was added and

the mix was incubated for 30 min at room temperature. Fresh complete growth media were added to cells, and the transfection mix was added dropwise. The cells were harvested 24–48 h later depending on the experimental goals.

**Plasmids**. C-terminal Flag-tagged mouse ARF6 and ARF1 were obtained from Addgene (Plasmids #52407 and #52402). The plasmids were then used to generate ARF6 G2A, G2A/K3R, K3R, Q67L, Q67L/K3R, T27N, T27N/K3R, ARF1 G2A, and ARF1 G2A/N3K by quick change mutagenesis. To generate NMT constructs for transient mammalian expression, HA-tagged human NMT1 and NMT2 were inserted into the pCMV-Tag-4a vector. mCherry-TfR-20 was a gift from Michael Davidson (Addgene plasmid # 55144; http://n2t.net/addgene: 55144; RRID: Addgene 55144). SIRT2 and Flag-SIRT2 constructs are previously reported[2]. The primer sequences for mutagenesis and cloning are provided in Supplementary Table 2.

**NP-40 lysis buffer**. In all, 25 mM Tris-HCl pH 7.4, 150 mM NaCl, 10% glycerol, and 1% Nonidet P-40, and protease and phosphatase inhibitors were added freshly.

**IP wash buffer**. IP wash buffer contained 25 mM Tris-HCl pH 7.4, 150 mM NaCl, and 0.2% Nonidet P-40.

**4% sodium dodecyl sulfate (SDS) lysis buffer**. SDS 4% lysis buffer contained 50 mM trimethylamine, 150 mM NaCl, and 4% SDS.

**ALK12 labeling to detect fatty acylation of ARF6 and ARF1 in cells**. ARF6 and ARF1 WT and mutants were transfected into HEK293T cells using PEI or FuGene 6 transfection reagent. NMT1/2 were transfected in 1:2 ratio of ARF6:NMT2. After 24 h, the cells were treated with 50–100 μM Alk12 for 7–24 h along with TM, DDD85646, or NAM as indicated in the figures. The cells were washed with cold phosphate-buffered saline (PBS), scraped, and collected at $500 \times g$ for 5 min, then were lysed in NP-40 lysis buffer with protease inhibitor cocktail for 30 min on ice with brief vortexing every 10 min. The lysates were incubated with anti-FLAG affinity beads at 4 °C for 2 h. The affinity beads were then washed three times IP and then re-suspended in 20 μl of IP washing buffer. The click chemistry reaction was performed by adding the following reagents: TAMRA azide (1 μl of 2 mM solution in dimethyl sulfoxide (DMSO)), TBTA (1 μl of 10 mM solution in dimethylformamide (DMF)), CuSO$_4$ (1 μl of 40 mM solution in H$_2$O), and TCEP (1 μl of 40 mM solution in H$_2$O). The reaction was allowed to proceed at room temperature for 30–60 min. Then SDS protein loading dye was added to 2× final concentration and the beads were heated at 95 °C for 10 min. After centrifugation at $17,000 \times g$ for 2 min, the supernatant was collected and treated with 300 mM hydroxylamine at 95 °C for 7 min. One microliter of each sample was diluted into 30 μl of 1× sample loading dye and 5–10 μl were analyzed by WB. In-gel fluorescence was detected with Typhoon FLA7000 (GE Healthcare Life Sciences). Protein loading was further analyzed by WB.

**SDS-polyacrylamide gel electrophoresis (PAGE) and WB analysis**. Running buffer: 30.3 g of Tris base, 144 g of Glycine, and 10 g of SDS were dissolved in 1 L of water to obtain 10× stock that was diluted in water to 1× before use.

Transfer buffer: 38 g of Tris base, 180 g of Glycine, and 6.25 g of SDS were dissolved in 1 L of water for a 10× stock that was further diluted to 1× in 20% methanol in water before use.

Blocking buffer: 5% bovine serum albumin (BSA) in 0.1%Tween20-PBS. Wash buffer: 0.1%Tween20-PBS.

6× SDS loading dye: 1.2 g of SDS, 6 mg of Bromophenol Blue, 4.7 ml of glycerol, and 1.2 ml of 0.5 M Tris (pH 6.8) were dissolved in 4.1 ml of water and 0.93 g of dithiothreitol (DTT) was added and dissolved. The dye was frozen in aliquots for further use.

The samples were denatured in the SDS loading dye and were loaded on a 12% polyacrylamide gel and were then resolved at 200 V for 1 h in running buffer. The proteins from the gel were transferred onto a polyvinylidene fluoride membrane activated in methanol for 30 sec at 330 mAmp for 1–2 h in cool transfer buffer. The membrane was then briefly rinsed with wash buffer and was blocked in the blocking buffer for 1 h at room temperature. The primary antibody was diluted in the blocking buffer, then was added to the membrane for an overnight incubation at 4 °C. The membrane was then washed three times with the wash buffer and was incubated with the secondary antibody (diluted in the blocking buffer) for 1 h at room temperature. The membrane was then washed three times and the signals were detected with Typhoon FLA7000 (GE Healthcare Life Sciences) or ChemiDoc MP (Bio-Rad) imagers after ECL plus application. All membrane incubation steps were done on a shaker.

**Co-IP to detect SIRT2–ARF6 interaction**. Flag-SIRT2 or Flag-ARF6 were transiently overexpressed in three 10-cm plates of 50% confluent HEK293T cells. The cells were collected 24 h later and were washed and lysed in 500 μl of NP-40 lysis buffer containing protease inhibitors for 30 min on ice with brief vortexing every 10 min. The lysates were spun down. The supernatants were combined with 500 μl of IP wash buffer (recipe above) and 20 μl of Flag beads and were subjected to 2 h

anti-FLAG-affinity IP at 4 °C. The FLAG beads were washed three times with cold IP wash buffer and were boiled for 10 min in 15 μl of 2× SDS protein loading dye. After boiling, the beads were vortexed and spun down for 2 min at $17,000 \times g$. The supernatants were analyzed by WB.

**Subcellular fractionation**. Four million HEK293T cells were seeded into 10-cm plates. Flag-ARF6 and described mutants (1 μg) and HA-NMT2 (3 μg) plasmids were transfected into the cells using the PEI reagent. After culturing for 24 h, 100 μM ALK12 was added and the cells were cultured for additional 15 h. They were then washed with cold PBS and were scraped into 500 μl of fractionation buffer (250 mM Sucrose, 20 mM HEPES pH 7.4, 10 mM KCl, 1.5 mM MgCl$_2$, 1 mM EDTA, 1 mM EGTA, freshly added 1 mM DTT, and protease inhibitor cocktail). The samples were incubated on ice for 15 min, after which they were passed through a 27-gauge needle 15 times and were left on ice for another 20 min. They were then centrifuged for 5 min at $900 \times g$ to pellet and remove the nuclei. The mitochondria was removed by another spin at $10,000 \times g$ for 5 min. Then membrane and cytosol fractions were separated by centrifugation at $17,000 \times g$ for 2.5 h. The membrane fraction was washed twice by resuspending in 500 μl of fractionation buffer and passing through a 27-gauge needle 10 times and then centrifuging at $17,000 \times g$ for 1 h between each wash. The membrane fractions were resuspended in 15 μl of 4% SDS lysis buffer (50 mM trimethylamine, 150 mM NaCl, 4% SDS) containing protease inhibitor cocktail followed by a dilution with 100 μl of 1% NP40 lysis buffer. Equal amounts of protein in the membrane and cytosolic fractions were subjected to FLAG IP by adding 15 μl of FLAG beads resuspended in 100 μl of IP wash buffer and rotating at 4 °C for 2 h. Click chemistry was performed as described above. In all, 10 μl of 2× SDS protein dye was added, and the samples were boiled for 7 min and were analyzed by SDS-PAGE and WB.

**SIRT2 reaction on myristoylated ARF6 peptides**. In all, 50 μM ARF6 peptides myristoylated on glycine 2 or lysine 3, 4 μM SIRT2, and 1 mM NAD$^+$ were added to 20 mM Tris pH 7.5 to a 50 μl volume. The reactions were rotated at 37 °C for 12 h and were quenched with 50 μl of acetonitrile (ACN). The samples were spun down at $17,000 \times g$ for 10 min, the supernatants were mixed with 5 μl of 50% trifluoroacetic acid (TFA) and were analyzed by LC-MS (LCQ Fleet from Thermo Scientific) with a binary gradient of 0.1% acetic acid in water and 0.1% acetic acid in ACN using Kinetex 5 μm EVO C18 100 Å, $30 \times 2.1$ mm LC column from Phenomenex.

**$^{32}$P-NAD$^+$ assay to detect lysine defatty acylation by SIRT2**. Flag-tagged ARF6 WT and ARF6 K3R were transiently transfected into SIRT2 KD HEK293T cells (five 10 cm plates per condition) using FuGene reagent and 5 μg of plasmid. After culturing for 24 h, 50 μM myristic acid was added and the cells were cultured overnight. The cells were washed with cold PBS and were lysed with NP-40 lysis buffer, and the expression of ARF6 and ARF6 K3R were determined by WB. ARF6 and ARF6 K3R were Flag-affinity purified from 10 mg of lysate using 40 μl of M2 Flag-agarose. Lysate without ARF6 or ARF6 K3R expression was also subjected to IP. The beads were split into two tubes for the $^{32}$P -NAD$^+$ assay. A reaction mixture containing 150 mM NaCl, 50 mM Tris, 10 mM DTT, and 2 μM SIRT2 or buffer only control was added to each reaction tube (10 μl each). Then 0.1 μCi of $^{32}$P-NAD$^+$ (final activity) was added to each tube, and the samples were briefly vortexed, spun down, transferred to 37 °C heat block, and incubated for 30 min with gentle tapping every 10 min. The samples were then spun down, and 2 μl of supernatant was spotted onto a TLC plate. The spots were dried and the products were resolved in a TLC chamber containing 0.75 M ammonium acetate (pH 7.1) and 70% ethanol. Then the plate was dried and exposed to the phosphor imaging screen (GE Healthcare, Piscataway, NJ) for 6–12 h. The phosphorescence was detected using Typhoon FLA7000. The peptide assays were performed in the same manner using 40 μM peptides.

**NMT reaction to detect the modification of ARF6 protein**. HEK293T cells were transiently transfected with ARF6 WT, K3R, G2A, and G2A/K3R (two 10-cm plates/construct). Twenty-four hours later, the cells were treated with 5 μM NMT inhibitor DDD85646 for 6 h. The cells were lysed in NP-40 lysis buffer (25 mM Tris-HCl pH 7.4, 150 mM NaCl, 10% glycerol, and 1% Nonidet P-40) containing the protease inhibitor cocktail, and the proteins were isolated by Flag-affinity beads purification as described above. After three washes, the beads were separated into three tubes and 100 μl of reaction mixture (50 mM Tris pH 8.0, 200 μM Alk12-CoA, and 10 μM NMT1/2 or PBS) were added to the beads. The reactions proceeded for 2 h at 30 °C while rotating. The beads were washed two times and click chemistry, in-gel fluorescence, and WB were performed as described above.

**Detection of ARF6 G2A lysine myristoylation by MS**. ARF6 G2A and NMT2 were transfected into 10 10-cm plates of SIRT2 KD HEK293T cells using FuGene 6 transfection reagent. After incubation for 24 h, the cells were treated with 50 μM myristic acid plus 7 mM NAM overnight. The cells were lysed with the NP-40 lysis buffer, and the protein was purified on Flag-affinity beads. MS was performed as previously described[2]. Briefly, the following conditions were used: LC gradient of

5–95% ACN with 0.1% formic acid (FA) from 0 to 160 min, 95–5% ACN with 0.1% FA from 160 to 161 min, 5% ACN with 0.1% FA from 161 to 200 min; flow rate: 0.25 μl/min; voltage applied to the nano-LC electrospray ionization source: 2.5 kV; orbitrap resolution: 120,000; MS1 scan range: $m/z$ 375–1575; MS2 scan range 112–867; ARF6 G2A myristoylation was identified by manual search.

**NMT reaction on ARF6 N-terminal-derived synthetic peptides**. Each 100 μl reaction contained 50 mM Tris (pH 8.0), 4–5 μM NMT1 or NMT2 or PBS, 200 μM myristoyl-CoA, and 100 μM peptide (added last). The mixtures were briefly vortexed and incubated at 30 °C for 2 h. The reactions were then quenched with 100 μl of ACN for 30 min. The samples were centrifuged for 10 min at $17,000 \times g$ and the supernatant was transferred to a new tube and analyzed by LC-MS with a binary gradient of 0.1% acetic acid in water and 0.1% acetic acid in ACN over 12 or 24 min. MS1 scan resolution was 150–2000 $m/z$ and MS2 (Supplementary Fig. 2) was 200–2000 $m/z$; for the precursor ion 744.5 $m/z$ (Iso. width 1.3 $m/z$). The reactions in Fig. 1d were separated on a binary gradient of 0.1% TFA in water and 0.1% TFA in ACN on analytical HPLC (Shimadzu UFLC).

**NMT kinetics**. For each reaction, 1 μM NMT1/2 and 100 μM myristoyl-CoA were added to 50 mM Tris (pH 8.0). After brief vortexing, the solution was transferred to reaction tubes in 96 μl aliquots. Then 4 μl of peptides dissolved in DMSO were added to each tube to final concentrations ranging from 200 μM to 3.13 μM. The reactions were rotated at 30 °C for 1 h (G2A peptide) or 0.5 h (K3R peptide) followed by quenching with 100 μl of ACN, vortexing, and incubating for 30 min at room temperature. The samples were then spun down for 10 min at $17,000 \times g$. The supernatants (190 μl) were transferred to new tubes and mixed with 10 μl of 50% TFA. The samples were analyzed by analytical HPLC (Shimadzu UFLC) using a binary gradient of 0.1% TFA in water and 0.1% TFA in ACN using an LC column Kinetex 5 μm EVO C18 100 Å, 150 × 4.6 mm from Phenomenex. Percentage of conversion was determined using the LabSolutions software. The reactions were performed in duplicates. Michaelis–Menten kinetic parameters were calculated using the GraphPad Prism 5 software.

**GTP loading analysis of overexpressed ARF6 WT and K3R**. ARF6 WT and K3R were transfected into HEK293T cells in 6 well plates. After culturing for 18 h, the cells were treated with 100 μM myristic acid and 25 μM TM to promote acylation for 24 h. The cells were washed with cold PBS and were lysed with 200 μl of the NP-40 lysis buffer. The GGA3-PBD (15 μl per sample) or protein A/G beads (30 μl per sample) were washed 3 times with 1 ml of IP wash buffer, added to the lysates in 200 μl of IP wash buffer and were rotated for 1 h at 4 °C. The beads were washed 3 times with 1 ml of the IP wash buffer, and after removing all the buffer with gel loading tips were boiled in 2× protein loading dye for 10 min, and then resolved by SDS-PAGE and analyzed by WB. For the analysis of the effects of NMT inhibition and NMT2 OE, the same procedure was followed except that NMT2 was co-transfected with WT ARF6 and 2 μM DDD85646 was used along with myristic acid treatment.

**GTP loading analysis of endogenous ARF6 by GGA3 pull down**. HEK293T cells were seeded in a 6-well plate: 200,000 cells per well. Twenty-four hours later, 1 ml of media with lentiviral particles containing shRNA against luciferase or SIRT2 were added, and after 24 h, the media were replaced with the normal growth media. Then another 24 h later, 100 μM of myristic acid was added and the GGA3 pull down was performed as described above immediately after 24 h.

**ARF6 T27N, T27N/K3R, Q67L, and Q67L/K3R colocalization with TfR**. Stable SIRT2 KD, NMT1 KD, NMT2 KD, or Luciferase KD HEK293T cells were plated in MatTek imaging dishes. The next day, cells were transfected with Flag-tagged ARF6T27N, T27N/K3R, Q67L, and Q67L/K3R. After 24 or 48 h, the cells were fixed with 4% paraformaldehyde (PFA)-PBS. The inhibitors (25 μM TM or 0.2 μM DDD85646) were added 24 h post-transfection, and the cells were incubated for an additional 24 h followed by fixation.

Immunofluorescence staining was performed as follows. The PFA-fixed cells were permeabilized and blocked with PBS containing 5% BSA and 0.1% Triton for 30 min. The samples were then incubated with the antibodies for TfR (CD-71, 1:200 dilution) overnight at 4 °C, then were washed twice with 0.1% Triton-PBS and incubated with rabbit anti-Flag antibody (1:1000) for 1 h at room temperature. After two washes, the samples were incubated with secondary antibodies (1:1000 dilution) goat anti-rabbit and anti-mouse conjugated to Cy3 or 647 or 488 fluorophores followed by three washes. All antibodies were diluted in the blocking buffer: PBS containing 5% BSA and 0.1% Triton. The samples were mounted with DAPI fluoromount-G (0100-20, Southern Biotech) and were analyzed on Zeiss 710 or Zeiss 880 confocal microscope using ×63 objective.

**Colocalization analysis of SIRT2 with endogenous TfR**. Four hundred thousand HEK293T cells were seeded into a well of a six-well plate. The next day, the cells were transfected with 1 μg of Flag-SIRT2 using PEI transfection reagent. Twenty-four hours later, the cells were trypsinized, and 500,000 cells were seeded into Mattek 35-mm glass bottom dishes. Twenty-four hours later, the cells were washed

with PBS and were fixed with 4% PFA-PBS. The staining was performed as described above using an overnight incubation at 4 °C with anti-TfR (CD-71) (sc-32272) Ab at 1:200 dilution and a rabbit anti-Flag tag Ab at 1:1000 dilution.

**Alk12-CoA synthesis**. To a solution of Alk12 (19 mg, 0.085 mmol) in THF (0.5 ml) at room temperature was added triethylamine (11.9 μl, 0.085 mmol) and then methyl chloroformate (6.6 μl). The resulting solution was stirred at room temperature for 1 h. This reaction solution was slowly added to a solution of acetyl coenzyme A litium salt (11 mg, 0.014 mmol) in aqueous 2.5% KHCO₃ (0.5 ml). The resulting solution was stirred at room temperature overnight (16 h). The reaction was quenched with 0.05 ml of acetic acid and extracted by dicholoromethane (3 × 2 ml). The organic layer was washed by water (5 ml). The combined aqueous layer was concentrated, and the crude product was purified by silica gel column (CHCl₃:CH₃OH:CH₃COOH:H₂O = 15:9:1:2) to afford product (8 mg, 58.8%) as white solid. The product was further purified by HPLC using a binary gradient of 0.1% TFA in water and 0.1% TFA in ACN. The synthesis scheme is provided in Supplementary Fig. 17.

**Full-length NMT expression and purification**. Human NMT1 and NMT2 were inserted into the pETHisTEV vector using the Gibson assembly strategy at NDE1 restriction site, and the plasmids were transformed into *Escherichia coli* BL21 (DE3) cells. A single colony was grown overnight in 30 ml of LB media with 50 μg/ml kanamycin. The culture was then added to 2 l of LB media with kanamycin and was grown until optical density 600 reached 0.6. In all, 0.2 mM IPTG was added to induce protein expression and the cultures were grown overnight at 18 °C. The cultures were spun down at 8000 RPM for 10 min, and the pellets were resuspended in 40 mM Tris pH 8.0, 400 mM NaCl, 10 mM imidazole, and 1 mM phenylmethylsulfonyl fluoride and were lysed by sonication. The lysates were spun down at $20,000 \times g$ for 30 min at 4 °C. The supernatants were then applied to the equilibrated His Trap affinity column (71-5027-68 AF), and the column was washed with 50 ml of 40 mM Tris pH 8.0, 400 mM NaCl, and 30 mM imidazole. The proteins were eluted with 40 mM Tris pH 8.0, 400 mM NaCl, and 200 mM imidazole and were concentrated to 2 ml using 30k filters (Amicon). They were further purified by size exclusion on a Superdex 75 16/600 column FPLC into PBS pH 7.4 or 20 mM Tris pH 7.5, 100 mM NaCl, and 1 mM DTT.

**SIRT2 expression and purification**. SIRT2 was expressed and purified from *E. coli* as previously described[51].

**Sample preparation for top–down MS**. Stable SIRT2 KD HEK293T cells were seeded in 20 15-cm plates to 50% confluency. Twenty-four hours later, each plate was transfected with 10 μg of ARF6-3XFlag (pJaff211 plasmid) and 5 μg of HA-NMT2. Twenty-four hours later, the cells were treated with 100 μM of myristic acid for 15 h. The cells were then washed with cold PBS, lysed with 10 ml of NP-40 lysis buffer and spun down for 20 min at 5000 RPM. 10 ml of IP wash buffer was added to the supernatant and Flag IP was performed for 4 h at 4 °C using 100 μl of beads. The beads were washed with the IP wash buffer 3 times followed by 5 washes with 1 ml of buffer containing 25 mM Tris pH 8.0 and 150 mM NaCl. The protein was eluted with 300 μl of 125 μM 3×FLAG peptide in 25 mM Tris and 150 mM NaCl buffer for 1 h two times. Sample was concentrated to 100 μl, and buffer exchange was performed by adding 300 μl of cold water and concentrating to 100 μl two times. Then 5 μl of ACN and 0.2 μl of formic acid were added to the sample. Protein yield was estimated at 30 μg.

**Top–down MS analysis**. For top–down MS, 20 μl of the sample described above was desalted using a C4 ZipTip (Millipore) and the eluent was diluted 1:5 with solvent A (5% ACN in water + 0.2% FA). Sample was injected onto a trap column (150 μm inner diameter (ID) × 3 cm) coupled with a nanobore analytical column (75 μm ID × 25 cm). The trap and analytical column were packed with polymeric reverse phase (PLRP-S, Agilent) media (5 μm, 1000 Å pore size). Samples were separated using a linear gradient of 10% solvent A to 45% solvent B (5% water, 95% ACN, 0.1% FA) over 35 min. MS data were obtained on a Q-Exactive HF (Thermo Fisher) mass spectrometer fitted with a custom nanospray ionization source. Intact MS data were obtained at a resolving power of 120,000 ($m/z$ 200). The top 2 $m/z$ species were isolated and fragmented using higher-energy collisional dissociation. Additional MS runs were completed targeting the specific ARF6 proteoforms of interest. Data were analyzed with both ProSightPC (Thermo Fisher) and ProSight Lite (CITE https://www.ncbi.nlm.nih.gov/pubmed/25828799) against a custom ARF6 database.

**Detection of endogenous ARF6 myristoylation**. Ten million of HEK293 cells (control or SIRT2 KD) were seeded in 15-cm plates (1 plate per condition). Twenty-four hours later, 100 μM ALK12 and 10 mM NAM were added for 15 h. The cells were collected and washed with cold PBS. They were then lysed with 1 ml of 4% SDS lysis buffer. In all, 1.8 mg of protein was brought to 700 μl and treated with 1 M hydroxylamine for 30 min at 37 °C. The protein was then precipitated with 2 volumes of methanol, 2/3 volumes of chloroform, and 4/3 volumes of water (all precipitation reagents were ice-cold). The samples were spun down at $4690 \times g$,

4 °C, for 30 min. The supernatants were discarded and the protein pellets were washed with 10 ml of ice-cold methanol twice. The pellets were air dried and were resolubilized with 400 μl of 4% SDS lysis buffer by rotating at 37 °C and vortexing until dissolved. Eight hundred micrograms of protein was brought up to 400 μl with 4% SDS lysis buffer and click chemistry was performed by adding 5 mM azide–PEG3–biotin conjugate (Sigma-Aldrich catalog number 762024) in water to 200 μM, 10 mM TBTA in DMF to 400 μM, 40 mM TCEP in water to 2 mM, and 40 mM CuSO4 in water to 2 mM final concentrations. The samples were incubated for 1 h at 30 °C while rotating. The proteins were precipitated as above, and the pellets were resolubilized in 200 μl of 4% SDS lysis buffer. Four hundred and thirty micrograms of protein was brought up to 200 μl with 4% SDS lysis buffer, and the samples were brought up to 1.4 ml with BriJ buffer (50 mM TEA pH 7.4, 150 mM NaCl, 1% BriJ v/v). Twenty microliters per sample of high-capacity streptavidin-conjugated beads were washed 3 times with 1 ml of BriJ buffer and were resuspended in the appropriate volume of BriJ buffer to add 100 μl to each sample. The samples were rotated at 4 °C overnight. The beads were washed 3 times with 1 ml of BriJ buffer. After the last wash, all of the buffer was removed, and the beads were boiled in 15 μl of 2× SDS sample loading dye for 10 min. The samples were analyzed by SDS-PAGE and WB. The signal was detected with ChemiDoc MP or Typhoon FLA7000 (when ChemiDoc was not sensitive enough to detect the input signal) scanners.

**In vitro SIRT2 reaction to remove ARF6 ALK12 labeling**. One million of HEK293T cells was seeded in 6-cm plates. One microgram of Flag-ARF6 WT or G2A plasmids were co-transfected with 3 μg of HA-NMT2 using PEI transfection reagent. Twenty-four hours later, 100 μM ALK12 was added and the cells were treated for 15 h. The cells were lysed with 1% NP40 lysis buffer and subjected to flag IP as described above. The washed beads were divided into three tubes for the SIRT2 reaction. The reaction was performed in 15 μl of 50 mM Tris pH 8.0, 100 mM NaCl, 2 mM MgCl2, 1 mM DTT, 1 mM NAD+, and 5 μM SIRT2. The reaction components except SIRT2 were combined and added to the beads, then SIRT2 was added to start the reaction. The samples were incubated for 1 h at 37 °C with gentle agitation. The beads were washed three times with IP wash buffer and were boiled with 10 min in 2× SDS sample loading dye. They were then analyzed by WB and in-gel fluorescence.

**NMT2 co-IP with ARF6 Q67L and ARF6 T27N**. One million HEK293T cells were seeded into 6-cm plates. Twenty-four hours later, the cells were transfected with the Flag-ARF6 mutant (1 μg), HA-NMT2 (2 μg), and empty vector (2 μg) plasmids. Twenty-four hours later, the cells were washed with cold PBS and were lysed with 300 μl of NP40 lysis buffer. Three hundred micrograms of total protein were subjected to Flag or HA IP by adding 300 μl of IP wash buffer containing 15 μl of beads and incubating for 2 hr at 4 °C. The beads were washed three times, and the proteins were eluted by boiling in 2× SDS sample loading dye. The samples were analyzed by WB.

**Detection of NMT-catalyzed single lysine myristoylation on ARF6 protein**. In vitro NMT reaction with ALK12-CoA on ARF6 protein was performed as described above. The beads were then washed three times with the IP wash, and each sample was split into two tubes. Then the SIRT2 reaction was performed as described above.

**Detecting the effect of ARF6 lysine myristoylation on pERK**. Three hundred and fifty thousand HEK293T cells (SIRT2 KD or Luciferase KD control) were seeded in each well of six-well plates. Twenty-four hours later, the cells were transfected with 0.5 μg of Flag-ARF6 WT or K3R and 1.5 μg of HA-NMT2 or empty vector (to achieve equal levels of ARF6 across conditions) using PEI transfection reagent. Twenty-four hours later, the cells were washed with serum-free media once and cultured in 1 ml of serum-free media for 15 h. The cells were scraped into the media, spun down for 5 min at 500 g, washed with 1 ml of ice-cold PBS, and spun down again. Immediately, the pellets were lysed with 4% SDS lysis buffer containing protease and phosphatase inhibitors and nuclease or were frozen at −80 °C for later lysis. Twenty micrograms of lysates were analyzed for pERK and 10 μg for total ERK by WB. High NMT OE can potentially also inhibit pERK by disrupting the ARF6 GTPase cycle through excessive lysine myristoylation.

**Detecting the effect of TM and DDD85646 on pERK**. Four hundred thousand HEK293T cells were seeded in each well of six-well plates. Twenty-four hours later, inhibitors were added at the indicated concentrations. Fifteen hours later, the cells were harvested and analyzed as described for "Detecting the effect of ARF6 lysine myristoylation on pERK".

**NMT catalytic domain cloning, expression, and purification**. NMT1 (115–496) and NMT2 (115–496) were each cloned into a pETHisTEV vector with a His6 tag and a HRV 3C PreScission protease cleavage site. The proteins were purified from clarified BL21 cell lysate using Ni-NTA resin as described above for full-length NMT; concentrated and buffer exchanged with a 10DG column; treated with

PreScission protease overnight at 4 °C; run back over Ni-NTA resin to remove tag, protease, or contaminants; and finally purified by size exclusion on a Superdex 75 16/600 column in 25 mM Tris-HCl pH 7.5, 120 mM NaCl, 1 mM DTT, and 1 mM MgCl2.

**Crystallization, data collection, and structure solution**. NMT2 was used at a final concentration of 8 mg/ml in a buffer of 25 mM Tris-HCl pH 7.5, 120 mM NaCl, 1 mM DTT, and 1 mM MgCl2. Before crystallization, 0.35 mM myristoyl-CoA and 3.5 mM KVLSKIF peptide were added and incubated at room temperature for 10 min. Protein solution was mixed in a 1:1 ratio with a well solution of 22% PEG 8000 and 0.1 M Bis Tris pH 6.5. Rod-shaped crystals grew in 3 weeks by the hanging drop vapor diffusion method. NMT1, 7.5 mg/ml in 25 mM Tris-HCl pH 7.5 and 120 mM NaCl, co-crystallized in the presence of 0.3 mM myristoyl-CoA and 0.3 mM AcKVLSKIF using a well solution of 18% PEG 8000, 100 mM NaCl, 100 mM sodium citrate pH 5.6, and 100 mM MgCl2.

Crystals were cryo-protected with 25% glycerol added to well solution and flash-frozen in liquid nitrogen. Datasets were collected at the NE-CAT beamline 24-ID-E at the Advanced Photon Source (Supplementary Table 1). Images were indexed, integrated, and merged using XDS and Aimless in the RAPD pipeline at NE-CAT and further cut according to CC1/2 and I/sigmaI statistics in Phenix[52]. Structures were solved using PHASER molecular replacement[53]. Models were constructed using iterative building in COOT[54] and refinement in Phenix[55]. Omit maps were calculated by removing the CoA, peptide, and myristoyl groups followed by simulated annealing refinement.

**Generation of SIRT1/2/3/6/7, NMT1/2 and HDAC11 KD HEK293T cell lines**. Packaging and shRNA plasmids for lentivirus generation were from Sigma. Luciferase targeting was used as negative control. Lentiviral particles were generated by co-transfection of shRNA plasmids with the packaging plasmids pCMV-dR8.2 and pMD2.G into HEK293T cells. The medium containing the lentivirus was collected 24 and 48 h after transfection, was filtered with 0.45 μM filters, and was later used to transduce cells. Two hundred thousand of HEK293T cells were seeded in six-well plates. One milliliter of media of lentiviral particles contained the following shRNA: SIRT1-1 TRCN0000218734 and SIRT1-2 TRCN0000229630; SIRT3-1 TRCN0000038893 and SIRT3-2 TRCN0000298766; SIRT7-1 TRCN0000359663and SIRT7-2 TRCN0000359594; SIRT6-1 TRCN0000378253 and SIRT6-2 TRCN0000232528; HDAC11-1 CCGGGTTTCTGTTTGAGCGTGTGGACTCGAGTCCACACGCTC AAACA GAAACTTTTTG and HDAC11-2 CCGGGCGCTATCTTAATGAGCTCAAC TCGAGTTGAGCTCATTAAGATAGCGCTTTTTG; Luciferase CCGGCGCTGAGTACTTCGAAATGTCCTCGAGGACATTTCGAAGTACTCA GCGTTTTTG; NMT1-1: TRCN0000035713 (CCGGGCCTGAGCAGAAA-TATGACCATCTCGAGATGGTCATATT TCTGCTCAGGTTTTTG); NMT1-2 TRCN0000289868 (CCGGCGGAAATTGG TTGGGTTCATTCTCGA GAATGAACCCAACCAATTTCCGTTTTTG); NMT2-1 TRCN0000291915 (CCGGCCAACGGTAAACTGACTGATTCTCGAGAATCAGTCAGTTTACCGT TGGTTTTTG); NMT2-2 TRCN0000303312 (CCGGGGAAATTGAAGTAGTC GATAATCTCGAGATTATCGAC TACTTCAATTTCTTTTTG); SIRT2-1: TRCN0000040219 (CCGGGCCATCTTTGAGATC AGCTATCTCGAGA TAGCTGATCTCAAAGATGGCTTTTTG), SIRT2-2: TRCN0000310335, (CCGGCCTGTGGCTAAGTAAACCATACTCGAGTATGGTTTACTTAGCCA CAGGTTTTTG).

The cells were selected with 2 μg/ml puromycin for 1 week.

**The analysis of KD efficiency by quantitative reverse transcription polymerase chain reaction (qRT-PCR)**. RNA was extracted from cells using the E.Z.N.A. Total RNA Kit I (Omega bio-tek, catalog # R6834-02). cDNA was synthesized using 0.5–1.0 μg of RNA using the SuperScript VILO cDNA Synthesis Kit (Invitrogen, catalog # 11754050) and was diluted to 100–200 μl with water. qRT-PCR was performed in 20 μl reactions containing 1 μl cDNA, 500 nM primers (NMT1: GGTCAGGGACCTGCCAAAAC, CATGGGTGTTCACCACTTCG; NMT2: TCCCAGCAAACATTCGGATTT, ACCCGTTTCGA TCTCAACTTCT; HDAC11: CACGCTCGCCATCAAGTTTC, GAAGTCTCGCTCATGCCCATT; glyceraldehyde 3-phosphate dehydrogenase (GAPDH): ACAACTTTGG TATCGTGGAAGG, GCCATCACGCCACAGTTTC), and SYBR Green (Thermo Fisher Scientific, ref. 4367659). Target gene expression was normalized to the levels of GAPDH.

**Colocalization analysis**. Colocalization was analyzed in Fiji using the JACop plugin. Pearson's coefficients with standard error of the mean are reported.

**Statistical analysis**. All statistics were obtained using unpaired two-tailed *t* test in the GraphPad Prism 5 or 6 software.

**Reporting summary**. Further information on research design is available in the Nature Research Reporting Summary linked to this article.

## Data availability

The source data underlying Figs. 1e, 2b–d, 3a, d, 4a, b, d–h, 5a–d, 6b–e, and 7a–e and Supplementary Figs. 1c–e, 7a, b, 8b, c, 9b, 10, 11a, 12, 13a–c, 14, and 16 are provided as a Source Data file. Atomic coordinates and structure factors for NMT2 and NMT1 have been deposited in the PDB under accession numbers 6PAU and 6PAV, respectively. Other data are available from the corresponding author upon request.

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

## Acknowledgements

This work was supported by NIH/NIDDK (DK107868), NIH/NIGMS (GM098621), HHMI, and NSF GRFP awards. We thank the National Resource for Translational and

Developmental Proteomics (supported by NIH P41 GM108569) for help with ARF6 top–down mass spectrometry, Cornell Proteomic and MS Facility for help with ARF6 G2A mass spectrometry, and Jun Young Hong for help with instruments and reagents. Imaging experiments were performed at Cornell BRC-Imaging facility (supported by NIH S10RR025502, NYSTEM C029155 grants), with support of Johanna M. Dela Cruz. ARF6 (pJAF215) and ARF1 (pJAF211) plasmids were a gift from Dr. Gregory Pazour, and mCherry-TFR-20 plasmid was a gift from Dr. Michael Davidson. We are grateful to Maurine Linder, Richard A. Cerione, Benjamin D. Cosgrove, Hui Jing, and Arash Latifkar for helpful discussions and suggestions. This work made use of the Northeastern Collaborative Access Team beamlines, which are funded by the National Institute of General Medical Sciences from the National Institutes of Health (P30 GM124165). The Eiger 16M detector on 24-ID-E beam line is funded by an NIH-ORIP HEI grant (S10OD021527). This research used resources of the Advanced Photon Source, a U.S. Department of Energy (DOE) Office of Science User Facility operated for the DOE Office of Science by Argonne National Laboratory under Contract No. DE-AC02-06CH11357. Preliminary X-ray crystallography experiments were also performed at the Advanced Light Source (Berkeley Center for Structural Biology, Lawrence Berkeley National Laboratory, DOE Contract No. DE-AC02-05CH11231 and NIH award P30 GM124169) and at the Cornell High Energy Synchrotron Source (CHESS, NSF award DMR-1829070), using the Macromolecular Diffraction at CHESS (MacCHESS, NIH Award GM-124166) facility. This material is based upon work supported by the National Science Foundation Graduate Research Fellowship Program under Grant No. DGE-1650441. Any opinions, findings, and conclusions or recommendations expressed in this material are those of the authors and do not necessarily reflect the views of National Science Foundation.

## Author contributions

T.K. designed and performed all the studies except those noted below. I.R.P. and S.L.H. performed crystallography and associated expression and purification of the NMT catalytic domains. I.R.P. performed structure solution and building and contributed the figures and text describing the crystallographic findings to the manuscript. S.Z., I.R.P., and T.K. purified full-length NMT1 and NMT2 and I.R.P. and X.Z. purified SIRT2 for in vitro studies. X.Z. performed mass spectrometry for ARF6 G2A. C.Z. synthesized peptides that were not purchased from Biomatik. K.N.J. and G.P.K. replicated key experiments. M.Y. synthesized ALK12-CoA and TM. C.J.D., P.M.T., and N.L.K. analyzed the ARF6 protein sample by top–down mass spectrometry. T.K. wrote and H.L. revised the manuscript with the input from all authors. H.L. directed the biochemical studies and J.C.F. directed the X-ray crystallography studies.

## Competing interests

Cornell University has patents on the SIRT2 inhibitor TM with H.L. as an inventor. All other authors declare no competing interests.
