## [Peer Review File · Nature Communications]

Reviewers' comments:

Reviewer #1 (Remarks to the Author):

The manuscript "NMT1 and NMT2 are Lysine myristoyltransferases Regulating the ARF6 GTPase Cycle" by Kosciuk et al. reports on an unexpected product formation and posttranslational modification (PTM) catalyzed by the enzyme N-myristoyltransferase (NMT).

PTMs diversify proteins, leading to specific changes in activity, stability, localization, etc. Lipidation of proteins is an important PTM that affects the association of modified proteins with intracellular membranes. For example, in N-myristoylation, a myristoyl moiety is transferred to the N-terminal amino group of proteins containing the GxxxS/T recognition motif, thereby increasing the affinity of the modified protein for a biological membrane. The reaction is catalyzed by NMTs.

In this paper, the group reports on a novel product formation mediated by NMTs: The enzyme cannot only transfer a myristoyl moiety to the N-terminal glycine, but also can modify the epsilon-amino group of lysines that succeed the glycine in the amino acid sequence. This modification is shown for the small GTPase ARF6, which contains such a recognition motif.

The authors combine in vitro and in cellulo analyses to verify that indeed the lysine side chain is myristoylated and also get further support from the NMT crystal structures. Interestingly, ARF6 can be myristoylated at both amino acids simultaneously thereby equipping the protein with high hydrophobicity leading to effective membrane localization. This finding may explain the previous observation why ARF6 – in contrast to other ARF-proteins – is always associated with membranes and not localized to the cytosol.

The work presented by the authors is conclusive and well presented. Indeed, it is interesting to see that myristoylations may occur at previously unrecognized amino acid positions and may be subject to reversible cleavage. It will be interesting to see whether this double modification of proteins also occurs on other targets. The conclusions of the experiments in vitro and in cellulo are conclusive and thus the paper deserves publication in Nature communications.

Minor points

- Briefly explain the ARF-activity cycle and its regulation in the introductory section.
- Figure 1i is mislabeled in the legend.
- please explain the occurrence of double bands for the NMT preparations. (e.g. Figure 2c,e)
- please provide the columns and resins used for analytical or preparative chromatography (e.g. in the methods section, the column details are missing for the "NMT2 kinetics").
- please also provide the suppliers of materials, hardware, software in the method section.
- please provide a reference for the statement "NMT is thought to predominantly act cotranslationally, however there is mounting evidence for its posttranslational activity". (page 5)
- Figure 4d,e, 5a appear to be processed by image preparation software too much. Please check.

Reviewer #2 (Remarks to the Author):

It has been known for many years that ARF-family GTPases are myristoylated at their N-termini on glycine-2 by N-myristoyltransferases. Here the authors show that the same NMTs (NMT-1 and

NMT-2) also catalyze myristoylation of one ARF isoform, ARF6, on an adjacent lysine residue. ARF6 is unique among the 6 mammalian ARFs in that it remains membrane-anchored even in its GDP-bound state, and the authors provide evidence that myristoylation of R3 is responsible for this phenomenon.

There are several important aspects of this study. First, it is the first identification of a lysine myristoyltransferase; both NMT1 and NMT2 appear to possess both glycine- and lysine myristoyltransferase activity. Second, it explains the anomalous behavior of ARF6 relative to the other ARFs, which has been a long-standing mystery in the field. Third, it identifies SIRT2 as an enzyme that selectively removes the myristate from lysine, without affecting myristoylation at G2. The authors propose a model in which R3 myristoylation allows ARF6 to remain membrane bound during its endocytosis and trafficking to recycling endosomes, but that removal of the myristate by SIRT2 is required for GTP loading of ARF6 at the endosomal compartment. This model is adequately supported by the data and will be of substantial interest to the field.

There are, however, several issues that need to be addressed:

1. Myristoylation of ARFs at G2 is thought to occur post-translationally, yet the authors' model requires that myristoylation of R3 occurs in non-ER compartments as part of the GTPase cycle. The evidence for this is somewhat circumstantial – that overexpressed NMT2 concentrates at the plasma membrane in cells co-expressing a constitutively active ARF6 mutant. Presumably NMTs associate with nucleotide-free ARF6 during its translation – why wouldn't R3 myristoylation occur simultaneously? Could this be measured using *in vitro* translation on microsomes?
2. The data suggesting that SIRT2 acts preferentially at recycling endosomes is also weak. The colocalization is not convincing, at least from the images shown. While this interpretation may be correct, more definitive proof would require higher magnification images from cells that are less crowded and better spread, and quantification of colocalization (e.g. Pearson's or Mander's coefficients, as used in Fig. S11).
3. Most of the experiments showing di-myristoylation (G2+R3) of ARF6 use overexpression of NMT. Even under this condition, it appears that the fraction of di-myristoylated protein is relatively small. What fraction of active, GTP-bound ARF6 is di-myristoylated in the absence of overexpressed NMT? This could be determined by pulldown using GGA3 and quantification of the bands representing mono- and di-myristoylated forms.

Reviewer #3 (Remarks to the Author):

Kosciuk and coauthors report a tour-de-force study of NMT1 and NMT2 involvement in lysine myristoylation on the ARF6 GTPase and the impact this modification has on regulation of the ARF6-GTPase cycle. This work exhaustively examines these questions in peptide-, protein-, and cell-based studies complemented by crystallographic studies of peptide complexes with NMT1 and NMT2 and mass spectrometry of full-length ARF6 which argues for dimyristoylation of this protein within the cell. The unanticipated involvement of NMTs in modification of lysine residues suggests a much broader role for these acyltransferases than previously thought. In both execution and subject, this study is sufficiently impactful to deserve consideration for publication in *Nature Communications*.

There are a number of specific issues that should be addressed in the manuscript detailed below, but in general the figure captions often contain insufficient detail to allow the reader to easily understand and interpret the data provided. For example, the subfigure caption for Figure 4a (a western blot figure with 6 separate immunoblots and 12 lanes) is "SIRT2 demyristoylates ARF6 in HEK293T cells" without any explanation of abbreviations, including those encountered for the first time in this figure. While the figure can be reasoned out, more descriptive captions would greatly enhance the readability of the manuscript.

- 1) NMT knockdown studies (Figure 2): Was the knockdown of NMT2 confirmed only by Western blot? The band density in Fig 2 Panel C for NMT2 is not overly dark and the extent of the knockdown is hard to assess. Given that this data is used to propose that NMT1 is the primary myristoyltransferase in HEK293 cells (page 7), I recommend confirming both NMT1 and NMT2 knockdown by another method such as QPCR.
- 2) Sequence preference studies for NMT1 and NMT2: In the text on page 10 and in Figure 3A and B, the authors do not note in the text that the studies in Figure 3B are using peptides rather than cell-expressed proteins and metabolic labeling. In addition, the anti-FLAG western blot in Figure 2 A indicates no expression for the "G2A + 2A" construct in lanes 6, 10, and 14. In light of this, the authors cannot say that addition of two alanines block modification as there is no evidence for the substrate protein being stably expressed.
- 3) Bottom page 10: Rather than stating the selectivity data indicate that hydrophobic residues are preferred and noncharged can be tolerated, it would be more appropriate to state that charged residues are not accepted at this position as the discrimination between Ala, Asn, and Phe is less than 2-fold from wild type in most cases.
- 4) Assignment of SIRT2 as the "eraser" for lysine myristoylation: The data supporting SIRT2-catalyzed demyristoylation of ARF6 is solid, but without testing other sirtuins and metal-dependent HDACs it is premature to say that "...SIRT2 is the physiological eraser" (last sentence on page 13 and Figure 4I). While the SIRT2 knockdown data does support this sirtuin playing a major role in catalyzing ARF6 deacylation, the authors must explore the ability of the other HDACs and demonstrate clear superiority by SIRT2 to make the claim currently in the paper.
- 5) Colocalization studies: The authors state conclusions regarding protein colocalization in several different experiments in the text without describing the methodology used to evaluate degree of colocalization determined by fluorescence microscopy. There is one mention of Mander's overlap coefficient analysis in the caption for figure S11. However, measuring colocalization and correlation by Pearson correlation analysis is considered a better method and is generally used in the field (Adler and Parmryd, Cytometry Part A 2010 733-742). The authors should reanalyze the colocalization studies and report Pearson correlation coefficients with explicit mention of this methodology in the figure captions or text where appropriate.
- 6) Figure 6c: The authors report (n=1) for the data in this section. Reporting fractionation experiments based on a single trial raises questions of reproducibility, particularly with challenging techniques such as these. Another trial should be performed to confirm the findings reported.
- 7) Figure S12: It is unclear why lanes 3 and 5 on the Ctrl gel appear different, as per the gel labels/legend these lanes should be identical. The same issue is present in lanes 4&6 and 10&12 in the SIRT2 KD gel. I assume this is an issue of incorrect lane labeling, but needs to be explained or corrected.

Minor comments:

- 1) The term "ectopically expressed/expressing" is used several times throughout the manuscript. I would suggest using the more common terms "transiently transfected" or "stably transfected" (if appropriate).
- 2) When error bars are shown, please state that the error shown represents (std dev, SEM, etc).
- 3) There are several incomplete citations in the references (e.g. refs 1-3), please correct.
- 4) There are multiple typos and spelling errors in the Supp Info, please proofread carefully.

We would like to thank all the reviewers for their very positive comments. We have addressed their concerns as detailed below and we believe the manuscript has been significantly improved.

Reviewer #1

We thank Reviewer 1 for the positive comment: *“The work presented by the authors is conclusive and well presented. Indeed, it is interesting to see that myristoylations may occur at previously unrecognized amino acid positions and may be subject to reversible cleavage. It will be interesting to see whether this double modification of proteins also occurs on other targets. The conclusions of the experiments in vitro and in cellulo are conclusive and thus the paper deserves publication in Nature communications.”* Reviewer 1’s minor points are addressed as detailed below:

- Briefly explain the ARF-activity cycle and its regulation in the introductory section.

We added this information to the introduction.

- Figure 1i is mislabeled in the legend.

We corrected this typo.

- Please explain the occurrence of double bands for the NMT preparations. (e.g. Figure 2c,e)

Biological studies and computational analysis (from uniprot) suggest there are 5 isoforms for NMT1 and two for NMT2; furthermore these proteins can be phosphorylated and have predicted ubiquitination sites, which could lead to multiple WB bands, which is observed by blotting with the anti-NMT antibodies (Fig. 2C). The addition of the HA tag to the NMT sequences could result in a better separation of two bands that are very close otherwise (Fig. 2C). Both bands expressed with our constructs are recognized by the anti-NMT antibodies (Fig. 2C) giving confidence that they correspond to the NMT enzymes.

- Please provide the columns and resins used for analytical or preparative chromatography (e.g. in the methods section, the column details are missing for the “NMT2 kinetics”).

We added this information to the methods section.

- Please also provide the suppliers of materials, hardware, software in the method section.

We added this information to the methods section.

- Please provide a reference for the statement *“NMT is thought to predominantly act cotranslationally, however there is mounting evidence for its posttranslational activity”*. (page 5)

We added the following reference: Thinon E, *et al.* Global profiling of co- and post-translationally N-myristoylated proteomes in human cells. *Nat Commun* **5**, 4919 (2014).

- Figure 4d,e, 5a appear to be processed by image preparation software too much. Please check.

We replaced the images with ones that have less processing and included raw images in the source data file.

Reviewer #2

We thank Reviewer 2 for the very positive comments: *“It has been known for many years that ARF-family GTPases are myristoylated at their N-termini on glycine-2 by N-myristoyltransferases. Here the authors show that the same NMTs (NMT-1 and NMT-2) also catalyze myristoylation of one ARF isoform, ARF6, on an adjacent lysine residue. ARF6 is unique among the 6 mammalian ARFs in that it remains membrane-anchored even in its GDP-bound state, and the authors provide evidence that myristoylation of R3 is responsible for this phenomenon. There are several important aspects of this study. First, it is the first identification of a lysine myristoyltransferase; both NMT1 and NMT2 appear to possess both glycine- and lysine myristoyltransferase activity. Second, it explains the anomalous behavior of ARF6 relative to the other ARFs, which has been a long-standing mystery in the field. Third, it identifies SIRT2 as an enzyme that selectively removes the myristate from lysine, without affecting myristoylation at G2. The authors propose a model in which R3 myristoylation allows ARF6 to remain membrane bound during its endocytosis and trafficking to recycling endosomes, but that removal of the myristate by SIRT2 is required for GTP loading of ARF6 at the endosomal compartment. This model is adequately supported by the data and will be of substantial interest to the field.”*

The issues raised by Reviewers have been addressed as described below:

1. Myristoylation of ARFs at G2 is thought to occur post-translationally, yet the authors' model requires that myristoylation of R3 occurs in non-ER compartments as part of the GTPase cycle. The evidence for this is somewhat circumstantial – that overexpressed NMT2 concentrates at the plasma membrane in cells co-expressing a constitutively active ARF6 mutant. Presumably NMTs associate with nucleotide-free ARF6 during its translation – why wouldn't R3 myristoylation occur simultaneously? Could this be measured using in vitro translation on microsomes?

We proposed that K3 myristoylation is post-translational for two reasons. First, in vitro, G2 myristoylation is a few times more efficient than K3 myristoylation based on the kinetics data. Second, even if K3 myristoylation initially occurs co-translationally, because it can be later removed by SIRT2 and then re-added by NMT, this later re-addition in the cycle must be post-translationally. Whether initially K3 can be modified co-translationally would be interesting to investigate in the future, but it is fair to say that K3 myristoylation can occur post-translationally.

2. The data suggesting that SIRT2 acts preferentially at recycling endosomes is also weak. The colocalization is not convincing, at least from the images shown. While this interpretation may be correct, more definitive proof would require higher magnification images from cells that are less crowded and better spread, and quantification of colocalization (e.g. Pearson's or Mander's coefficients, as used in Fig. S11).

We have added a new image addressing these concerns and also did the Pearson's coefficient quantification. The data are shown Figure 6E. It should be pointed out that SIRT2 is also localized to other places, but it has significant localization at the recycling endosomes.

3. Most of the experiments showing di-myristoylation (G2+R3) of ARF6 use overexpression of NMT. Even under this condition, it appears that the fraction of di-myristoylated protein is relatively small. What fraction of active, GTP-bound ARF6 is di-myristoylated in the absence of overexpressed NMT? This could be determined by pulldown using GGA3 and quantification of the bands representing mono- and di-myristoylated forms.

This is a great suggestion, but unfortunately we were unable to resolve dimyristoylated ARF6 using the GGA3 pull down. This could be due to several technical limitations. To accumulate

dimyristoylated ARF6, SIRT2 has to be inhibited, which pushes ARF6 into inactive state making the GGA3 binding inefficient. Without SIRT2 inhibition or depletion, the abundance of modification might be below the detection limit. Fortunately, we were able to partially address this question: we determined the abundance of dimyristoylated ARF6-GTP by performing ALK12 labeling on ARF6 Q67L in HEK293T cells without NMT overexpression (Figure S17). While we found about 3% of dimyristoylated ARF6, it might be an underestimation for several reasons. First, endogenous ARF6 might have a greater NMT availability (compared to overexpressed ARF6). Second, comparing the fluorescence signal and the Flag WB signal, it is clear that click chemistry efficiency for the dimyristoylation product is much smaller, likely caused by the steric hindrance due to proximity of the two acyl chains. At the same time, we do not anticipate high levels of ARF6 lysine myristoylation at normal conditions. Because of the cycling nature of this modification and its coupling to GTP loading, the steady state level of this modification could be very low, but it still can play an important role for the GTP loading of ARF6. It will be of interest to explore the conditions at which this modification accumulates. Such conditions might include inhibition of translation under certain stress conditions. Without newly synthesized proteins that needs to be glycine myristoylated, NMT may be more available to act on its ARF6 K3. We are excited to explore such directions in the future.

Reviewer #3

We appreciate very much Reviewer 3's positive comments: "*Kosciuk and coauthors report a tour-de-force study of NMT1 and NMT2 involvement in lysine myristoylation on the ARF6 GTPase and the impact this modification has on regulation of the ARF6-GTPase cycle. This work exhaustively examines these questions in peptide-, protein-, and cell-based studies complemented by crystallographic studies of peptide complexes with NMT1 and NMT2 and mass spectrometry of full-length ARF6 which argues for dimyristoylation of this protein within the cell. The unanticipated involvement of NMTs in modification of lysine residues suggests a much broader role for these acyltransferases than previously thought. In both execution and subject, this study is sufficiently impactful to deserve consideration for publication in Nature Communications.*" The issues raised by Reviewer 3 are addressed as detailed below:

1. *In general the figure captions often contain insufficient detail to allow the reader to easily understand and interpret the data provided.*

We provided more information in figure captions.

2. *NMT knockdown studies (Figure 2): Was the knockdown of NMT2 confirmed only by Western blot? The band density in Fig 2 Panel C for NMT2 is not overly dark and the extent of the knockdown is hard to assess. Given that this data is used to propose that NMT1 is the primary myristoyltransferase in HEK293 cells (page 7), I recommend confirming both NMT1 and NMT2 knockdown by another method such as qPCR.*

We performed qRT-PCR and included the results in the Supplementary Figure 16. NMT knockdown is toxic to cells and we noticed that one of the NMT2 KD cell lines overcame the effect of shRNA by the time we performed qRT-PCR. However, the other KD cell line was still ok and the data is sufficient to conclude that endogenous NMT can regulate lysine myristoylation.

3. *Sequence preference studies for NMT1 and NMT2: In the text on page 10 and in Figure 3A and B, the authors do not note in the text that the studies in Figure 3B are using peptides rather than cell-expressed proteins and metabolic labeling. In addition, the anti-FLAG western blot in Figure 2 A indicates no expression for the “G2A + 2A” construct in lanes 6, 10, and 14. In light of this, the authors cannot say that addition of two alanines block modification as there is no evidence for the substrate protein being stably expressed.*

We made the discussion of 3B clearer and stated in the text and figure legend that the G2A+2A mutant could not be stably expressed.

4. *Bottom page 10: Rather than stating the selectivity data indicate that hydrophobic residues are preferred and noncharged can be tolerated, it would be more appropriate to state that charged residues are not accepted at this position as the discrimination between Ala, Asn, and Phe is less than 2-fold from wild type in most cases.*

We modified the statement in the text.

5. *Assignment of SIRT2 as the “eraser” for lysine myristoylation: The data supporting SIRT2-catalyzed demyristoylation of ARF6 is solid, but without testing other sirtuins and metal-dependent HDACs it is premature to say that “...SIRT2 is the physiological eraser” (last sentence on page 13 and Figure 4I). While the SIRT2 knockdown data does support this sirtuin playing a major role in catalyzing ARF6 deacylation, the authors must explore the ability of the other HDACs and demonstrate clear superiority by SIRT2 to make the claim currently in the paper.*

We have tested several other sirtuins and HDACs and the data (Supplementary Figure 15) showed that SIRT2 is the major deacylase for ARF6.

6. *Colocalization studies: The authors state conclusions regarding protein colocalization in several different experiments in the text without describing the methodology used to evaluate degree of colocalization determined by fluorescence microscopy. There is one mention of Mander’s overlap coefficient analysis in the caption for figure S11. However, measuring colocalization and correlation by Pearson correlation analysis is considered a better method and is generally used in the field (Adler and Parmryd, Cytometry Part A 2010 733-742). The authors should reanalyze the colocalization studies and report Pearson correlation coefficients with explicit mention of this methodology in the figure captions or text where appropriate.*

We reanalyzed the studies and reported the Pearson’s correlation coefficients.

7. *Figure 6c: The authors report (n=1) for the data in this section. Reporting fractionation experiments based on a single trial raises questions of reproducibility, particularly with challenging techniques such as these. Another trial should be performed to confirm the findings reported.*

The replicates are in Figure S14A and B.

8. *Figure S12: It is unclear why lanes 3 and 5 on the Ctrl gel appear different, as per the gel labels/legend these lanes should be identical. The same issue is present in lanes 4&6 and 10&12 in the SIRT2 KD gel. I assume this is an issue of incorrect lane labeling, but needs to be explained or corrected.*

The label indicating the K3R mutation was missing in the original figure making it confusing. We apologize for that. The corrected version is currently the Supplementary Figure 13.

9. *Minor comments:*

- 1) *The term “ectopically expressed/expressing” is used several times throughout the manuscript. I would suggest using the more common terms “transiently transfected” or “stably transfected” (if appropriate).*
- 2) *When error bars are shown, please state that the error shown represents (std dev, SEM, etc).*
- 3) *There are several incomplete citations in the references (e.g. refs 1-3), please correct.*
- 4) *There are multiple typos and spelling errors in the Supp Info, please proofread carefully.*

We made modifications to address all of these suggestions.

REVIEWERS' COMMENTS:

Reviewer #2 (Remarks to the Author):

The authors have done a good job of addressing my earlier concerns. I understand the technical difficulties in determining the fraction of dimyristoylated Arf6 that is active, and accept the authors' explanation for their inability to provide this information.

I have one minor issue that can be addressed by changes to the text: The authors refer to transferrin receptor-containing endosomes as recycling endosomes (Fig. 6E), but TfR is also present in early endosomes. Based on their location in the one image shown, it is possible that SIRT2 is also present on early endosomes as well as the ERC. A statement to this effect should be included in the text. Also, what do the points in Fig 6E represent? Are these individual cells or individual endosomes? This should be explained in the figure legend.

Reviewer #3 (Remarks to the Author):

With the added experimental controls, revised analysis of protein colocalization imaging data, and textual revisions the revised manuscript addresses the issues raised by all reviewers. I recommend publication of the revised manuscript.

Reviewer #2

The authors have done a good job of addressing my earlier concerns. I understand the technical difficulties in determining the fraction of dimyristoylated Arf6 that is active, and accept the authors' explanation for their inability to provide this information.

I have one minor issue that can be addressed by changes to the text: The authors refer to transferrin receptor-containing endosomes as recycling endosomes (Fig. 6E), but TfR is also present in early endosomes. Based on their location in the one image shown, it is possible that SIRT2 is also present on early endosomes as well as the ERC. A statement to this effect should be included in the text. Also, what do the points in Fig 6E represent? Are these individual cells or individual endosomes? This should be explained in the figure legend.

Response: Thank you for the comment. Indeed, TfR is also present in early endosomes and it is likely that SIRT2 might act on ARF6 in that compartment. We modified our conclusion and model accordingly. Furthermore, we replaced the data in Figure 6E with new data depicting colocalization of transiently overexpressed SIRT2 with endogenous TfR for the following reason. While waiting for the decision on the manuscript, we observed that overexpressed SIRT2 localized differently from the staining we obtained for endogenous SIRT2 using the Santa Cruz antibody that is reported to be suitable for immunofluorescence. This prompted us to validate the antibody. Unfortunately, the in-lab validation of the antibody in SIRT2 knockout mouse embryonic fibroblasts and SIRT2 knockdown and overexpression HEK293T cells revealed that the antibody does not target SIRT2. For that reason, we also removed Supplementary Figure 10 that showed endogenous SIRT2 colocalization with endocytosed fluorescent transferrin. Because of the antibody problem, we instead showed that transiently overexpressed SIRT2 colocalized with endogenous TfR with a Pearson's correlation coefficient of about 0.5 (Figure 6E). We also validated the TfR antibody by colocalization studies with transiently overexpressed TfR-mCherry and the antibody showed correct specificity. Each point in the quantification represents one cells, which is now explained in the figure legend.

Reviewer #3

With the added experimental controls, revised analysis of protein colocalization imaging data, and textual revisions the revised manuscript addresses the issues raised by all reviewers. I recommend publication of the revised manuscript.

Response: Thank you!